



# The Coupled Model Intercomparison Project (CMIP): Reviewing project history, evolution, infrastructure and implementation

Paul J. Durack[1], Karl E. Taylor[1], Peter J. Gleckler[1], Gerald A. Meehl[2], Bryan N. Lawrence[3], Curt Covey[1], Ronald J. Stouffer[4], Guillaume Levavasseur[5,6], Atef Ben-Nasser[5,7], Sebastien Denvil[8,a], Martina Stockhause[9], Jonathan M. Gregory[3,10], Martin Juckes[11,12], Sasha K. Ames[1], Fabrizio Antonio[13], David C. Bader[1], John P. Dunne[14], Daniel Ellis[15], Veronika Eyring[16,17], Sandro L. Fiore[18,b], Sylvie Joussaume[5,7], Philip Kershaw[19,12], Jean-Francois Lamarque[c], Michael Lautenschlager[d], Jiwoo Lee[1], Chris F. Mauzey[1], Matthew Mizielinski[10], Paola Nassisi[13], Alessandra Nuzzo[13], Eleanor O'Rourke[15], Jeffrey Painter[e], Gerald L. Potter[e], Sven Rodriguez[5,6], and Dean N. Williams[f]

[1]Program for Climate Model Diagnosis & Intercomparison (PCMDI), Lawrence Livermore National Laboratory (LLNL), Livermore, California, USA
[2]National Center for Atmospheric Research (NCAR), Boulder, Colorado, USA
[3]NCAS, Department of Meteorology, University of Reading, Reading, UK
[4]The University of Arizona (UA), Tucson, Arizona, USA
[5]Institut Pierre Simon Laplace (IPSL), Paris, France
[6]Sorbonne University (SU), Paris, France
[7]Centre National de Recherche Scientifique (CNRS), Paris, France
[8]European Centre for Medium-Range Weather Forecasts (ECMWF), Reading, UK
[9]German Climate Computing Center (DKRZ), Hamburg, Germany
[10]Met Office (MOHC), Exeter, UK
[11]Kellogg College, University of Oxford, Oxford, UK
[12]UK Research and Innovation (UKRI) Science & Technology Facilities Council (STFC), Oxfordshire, UK
[13]CMCC Foundation - Euro-Mediterranean Center on Climate Change (CMCC), Lecce, Italy
[14]Geophysical Fluid Dynamics Laboratory, National Oceanic & Atmospheric Administration (NOAA-GFDL), Princeton, New Jersey, USA
[15]CMIP International Project Office (CMIP-IPO); European Space Agency Centre for Space Applications & Telecommunications (ESA-ECSAT), Oxfordshire, UK
[16]Deutsches Zentrum für Luft- und Raumfahrt (DLR), Institut für Physik der Atmosphäre, Oberpfaffenhofen, Germany
[17]University of Bremen, Institute of Environmental Physics (IUP), Bremen, Germany
[18]University of Trento, Trento, Italy
[19]Centre for Environmental Data Analysis (CEDA), RAL Space, Oxfordshire, UK
[a]formerly at: Institut Pierre Simon Laplace (IPSL)
[b]formerly at: CMCC Foundation - Euro-Mediterranean Center on Climate Change (CMCC)
[c]formerly at: National Center for Atmospheric Research (NCAR)
[d]formerly at: German Climate Computing Center (DKRZ)
[e]formerly at: Program for Climate Model Diagnosis & Intercomparison (PCMDI)
[f]formerly at: Earth System Grid Federation (ESGF), Livermore, California, USA

**Correspondence**

Paul J. Durack (pauldurack@llnl.gov)





**Abstract.**

The CMIP6 project was the most expansive and ambitious Model Intercomparison Project (MIP), the latest in a long history, extending back four decades. CMIP has captivated and engaged a broad, growing international community focused on improving our climate understanding. It has anchored our ability to quantify and attribute the drivers and responses of the observed climate changes we are experiencing today.

The project's profound impact has been achieved by combining the latest climate science and technology. This has enabled the production of the latest-generation climate simulations and disseminating their output, which has seen increased community attention in every successive phase. The review emphasizes the pragmatics of progressively scaling up efforts, the evolution of how the MIPs were implemented, and the coordinated international efforts to establish a minimal infrastructure to make that possible, most recently delivering CMIP6.

**Keywords.** AMIP, CMIP, AMIP1, AMIP2, CMIP1, CMIP2, CMIP3, CMIP5, CMIP6, CMIP6Plus, climate, earth system, modelling, atmosphere, ocean, land, sea ice, land ice, model, intercomparison, science

# 1   Introduction

The Model Intercomparison Project (MIP) concept, a well-worn terminology in climate science, has existed for many decades. Among other things, the MIP era has delivered two critical advancements to climate science. The first is that modelling groups contributing to any MIP have agreed to make their model output available to be scrutinized by the research community. Before the advent of MIPs, this was generally not the case, with model analysis typically performed by individual modelling groups or close collaborators. The second, contributing to a MIP means agreeing to adhere to a defined experimental design, meaning simulation data could be directly compared.

Scientists, stakeholders, and policymakers often refer to the acronyms of the Coupled Model Intercomparison Project (CMIP) and Intergovernmental Panel on Climate Change (IPCC) interchangeably. However, this incorrectly reflects their independence, which is scientifically and organizationally separate. Historically, CMIP phases were implemented before IPCC phases, which assessed CMIP-enabled climate science. The MIPs further inspired international effort coordination, leading to considerable science collaboration that has enabled breakthroughs, including the building consensus that human influence is "*unequivocal*" in driving the observed climate changes we are experiencing today (see Figure 6; Eyring et al., 2021a). More comprehensive international coordination has marked their development as a way to systematically evaluate observed climate changes, generate plausible future projections driven by varying futures of human development, and assess the fidelity of global climate models (GCMs) and Earth System Models (ESMs) to reproduce past changes and gauge their responses and sensitivities to realistic and idealised climate forcing. The multi-model approach to climate change assessment is now routine, with documented and reproducible experimental protocols and input "forcing" datasets facilitating standardised simulations that can be directly compared across models and with observations to ascertain the externally-forced signal from noise and identify robust model responses versus outliers and anomalies that are not representative of our real world best estimates.



To provide context for the most recent phase (CMIP6; Eyring et al., 2016a) and the planning of the project's future, we discuss the efforts dating back to the MIP origins and their contribution to making climate community collaboration what it is today. Small teams or individuals initiated many of the steps taken. While evidence of this history is scattered in the literature, we attempt to bring them together to tell a coherent story of how CMIP got to where it is today. Our emphasis details the pragmatics of what it took to progressively scale up efforts, most recently delivering CMIP6. Thus, this review describes the evolution of how the MIPs were implemented and the coordinated efforts to establish a minimal infrastructure to make that possible.

In section 2, we discuss MIP phases from 1990 to the present. In section 3, we highlight the CMIP6 experimental design, and in section 4, the supporting organizations and infrastructure. We then describe CMIP6 supporting projects in section 5, section 6 attempts to quantify the project's impact, and section 7 summarises the state of CMIP6, its planned completion and ongoing progress. In section 8, we look forward to the upcoming CMIP7 phase and highlight ongoing activity development.

## 2 CMIP6 in context with the past

CMIP6 is the latest in a long history of internationally coordinated climate modelling intercomparison projects. The activities grew out of self-organizing communities that built collaborations around science questions and the recognition that systematic multi-model evaluation was needed to build confidence in climate models as effective tools for climate change prediction.

The scientific motivation for early studies can be linked to coordinated activities organized by the US Department of Energy (DoE), Carbon Dioxide Research Division. In the early 1980s, the DoE $CO_2$ Climate Research Plan was established (Riches, 1983), which laid out an ambitious proposal to model, detect, and observe $CO_2$-induced global and regional climate changes, building on a growing international body of climate science extending back to early seminal works (e.g., Arrhenius, 1896; Chamberlin, 1899; Charney et al., 1979). The program commissioned a set of six state-of-the-art reports (e.g., MacCracken and Luther, 1985) highlighting uncertainties in the current-generation atmospheric general circulation models (GCMs), their simplifications and parameterizations, well ahead of the Intergovernmental Panel on Climate Change (IPCC) program which was endorsed by the World Meteorological Organization (WMO) and United Nations Environment Programme (UNEP) in December 1988.

This led to a coordination of international efforts, first establishing the Intercomparison of Radiation Codes used in Climate Models project (ICRCCM) in 1982 (Luther et al., 1988; Ellingson and Fouquart, 1991), which evolved into a World Climate Research Program (WCRP) and International Radiation Commission joint working group in 1984. The early work highlighted enormous longwave clear sky discrepancies across codes, of order 30-70 W m$^{-2}$. It led to the recommendation that models needed to be validated against observations to prove their utility, and led to the follow-on Spectral Radiance Experiment (SPECTRE; Ellingson et al., 1990) targeted at collecting the necessary observations and co-sponsored by US National Aeronautics and Space Administration (NASA; Ellingson and Wiscombe, 1996). Over 22 participants submitted 29 calculation sets to the project, with 7 GCM configurations contributing (Ellingson et al., 2016). The observational design and much of the SPECTRE proposal were subsequently reused to establish the US DoE Atmospheric Radiation Measurement Program (ARM;



U.S. Department Of Energy, 1990), initiated in January 1989. Off the back of this early science momentum, in 1984, the DoE initiated the first international GCM intercomparison project to understand the significant differences across $CO_2$-forced

simulations at the time, the first MIP, FANGIO and its descendants AMIP and CMIP (see subsection 2.1; Ellingson et al., 2016).

These early projects and their evolution to the present day are described in the following sections.

## 2.1 Atmospheric Model Intercomparison Project phases (AMIP1/2)

Intercomparison of near-term weather, as simulated with different atmospheric models, occurred soon after the first global

models were developed in the 1950s (e.g., Gates, 1992b; Edwards, 2011). These activities became more coordinated in the early-1970s through the guidance of the Working Group on Numerical Experimentation (WGNE), which was formed in 1967 under the international Global Atmospheric Research Program (GARP), a precursor to the World Climate Research Programme (WCRP; Gates, 1992b). The first internationally coordinated climate model experimentation occurred in the late 1980s with the scientifically targeted Feedback ANalysis of GCMs and In Observations (FANGIO) project. FANGIO was explicitly designed

to look at feedback mechanisms, defining a sea surface temperature perturbation that prescribed a defined climate change. This project attracted contributions from 19 distinct atmospheric model configurations, and concluded that the cloud feedbacks across models was the major cause of the differences in modelled climate sensitivity (Cess and Potter, 1988; Cess et al., 1989, 1990; Cess and Hameed, 1991).

The nascent community progress quickly expanded into the first phase of the Atmospheric Model Intercomparison Project

(AMIP, hereafter referred to as AMIP1; Gates, 1992a), which was organized under the WGNE purview to help identify systematic errors, narrow the range of model results, and assist in prioritizing model development to reduce those errors. The acronym AGCM (Atmospheric General Circulation Model) was coined to define model configurations for AMIP, with many of the contributing model configurations also including simplified land-surface representations (e.g., Budyko, 1961; Manabe, 1969; Vargas Godoy et al., 2021).

AMIP1 provided a community AGCM experimental protocol with time-varying boundary conditions (later referred to as "forcing") of sea surface temperature (SST) and sea-ice for the 1979-1988 period, in addition to the recommended solar constant (1365 W m$^{-2}$) and carbon dioxide concentration (345 ppm; Gates, 1991). It augmented contributing atmospheric model configurations to 27, with 26 community "diagnostic subprojects" established to analyze the simulations (Gates, 1995).

The project expanded the required model output from simple global and zonal means for the perpetual July experiment

requested in FANGIO to monthly mean time series in three dimensions (120 covering the 10-year AMIP1 period). For many participating models, AMIP1 was the first opportunity to run their model longer than one annual cycle, with computer time provided by the Program for Climate Model Diagnosis and Intercomparison (PCMDI; see section 4).

Early AMIP1 results were assessed in the IPCC First Assessment Report (FAR; Gates et al., 1990). The successes of AMIP1, including the entrainment of a broader analysis community to assist in model evaluation, motivated the modelling community

to revisit the exercise to determine if and how models were improving. To do so, a second phase was established (AMIP2), including an expanded experimental design with improved boundary conditions (Liang et al., 1997; Taylor et al., 2000). The



AMIP2 requested model "Standard Output" (see Table 1 and subsection 4.3) was increased considerably beyond AMIP1, with selected higher frequency data to facilitate process-level analysis. In preparation for AMIP2, an AMIP Panel established by the WGNE reviewed 38 diagnostic subproject proposals (Gleckler, 2001), ultimately leading to comprehensive and systematic

model evaluation. As with AMIP1, the AMIP2 community analysis was assessed in the IPCC Second Assessment Report (SAR; Gates et al., 1996), with a focus on assessing mean errors and multi-model consistency across the archive (Gates, 1995).

The core AGCM experiment pioneered by AMIP1/2 is now a de facto modelling community benchmark experiment included in each subsequent phase (e.g., Table A1).

## 2.2   Coupled Model Intercomparison Project phases (CMIP1/2/2+)

In parallel, work continued at modelling centres to expand complexity to incorporate a dynamic ocean, amongst other climate system components, which had begun in the late 1960s and early 1970s (e.g., Manabe and Bryan, 1969; Bryan et al., 1975; Manabe et al., 1975). The AMIP1/2 AGCM experimental protocols, forced with fixed SST and sea-ice, could not simulate future (or past) climate changes. The recognition that climate change cannot be fully simulated without properly considering interactions with other major systems, such as the ocean, led to the establishment of the WCRP Steering Group on Global

Coupled Models (SGGCM) in November 1990 (Meehl, 2023).

Over the following years, the activity entrained more oceanographers working in parallel on ocean model development, encapsulated by work hosted by the WCRP CLImate VARiability and Predictability (CLIVAR) project (the CLIVAR Working Group on Ocean Model Development (WGOMD) was subsequently established at the second WCRP Working Group on Coupled Modelling (WGCM) meeting in Melbourne, 1998). The growing coordination led to the Coupled Model Intercom-

parison Project (CMIP) acronym being defined and the first phase, CMIP1, designed after a workshop at Scripps Institution of Oceanography in October 1994 (Meehl, 1995). The acronym AOGCM (Atmosphere-Ocean General Circulation Model) was coined to define model configurations for this project.

At the same time, the SGGCM established a small CMIP Panel responsible for planning and providing scientific direction for CMIP, and PCMDI committed to hosting the model output. CMIP1 was focused on collecting data and documenting

features of AOGCMs of present-day fixed climate, loosely following the AMIP1 protocol and available forcing data (pdcntrl; see Appendix A, Table A1). The new MIP data resource led to considerable community attention and model evaluation with the building archive (Lambert and Boer, 2001; Räisänen, 2001; Villwock and Mitchell, 2003).

In September 1995, the CLIVAR decadal-centennial variations and anthropogenic climate change (DecCen/ACC) Numerical Experimentation Group (NEG-2) was established. SGGCM transitioned into CLIVAR NEG-2, which defined an ambitious

program of modelling studies, including CMIP (Coughlan, 1996; Villwock, 1996). The project gathered momentum quickly, with a broad expansion in community-proposed diagnostic subprojects leveraging science insights from the growing archive, and a second phase, CMIP2, was planned by the CMIP Panel as part of CLIVAR NEG-2, and announced in January 1997 (Meehl et al., 1997, 2000). CMIP2 broadened the science remit, expanding the focus to include a pre-industrial ($\sim$1860) control (picntrl) and transient climate change experiments with $CO_2$ increasing at 1% per year (1pctCO2, see Table A1; Meehl

et al., 2003; Villwock and Mitchell, 2003).



In parallel to the CMIP growth, there was a recognition that coordinating this activity required a dedicated working group. In April 1997, the WGCM was created at a CLIVAR Scientific Steering Group (SSG) meeting in Washington DC, and the membership of CLIVAR NEG-2 transitioned to WGCM to meet the growing project needs (Detemmerman, 1997) this brought the project structure and hierarchy in line with what we have in place today.

As in AMIP, the CMIP model output received considerable attention with the resulting CMIP1 papers assessed, like AMIP, in the IPCC SAR (Gates et al., 1996) and IPCC Third Assessment Report (TAR; McAvaney et al., 2001).

With the growing interest in available model data, a recognition was made that the "standard output" collected to date was only a small subset of model data produced. For the preceding phases, for most variables, time-averaged quantities were collected. In contrast, the time-varying monthly mean information available for a handful of fields (surface temperature, pre-

cipitation, and sea level pressure) enabled far broader scientific investigations.

Recognizing this opportunity, an augmented phase CMIP2+ was identified and announced in May 2000 (Villwock and Mitchell, 2003; Meehl et al., 2003, 2005a), with a subset of CMIP1/2 contributing modelling groups augmenting their existing data contributions with considerably more variable and time coverage, including daily data, if available (AchutaRao et al., 2004).

CMIP2+ enabled research beyond the scope possible with CMIP1 and CMIP2, most notably a comprehensive appraisal of climate models for the US DoE (AchutaRao et al., 2004). However, it soon became apparent that contributors and users would greatly benefit from detailed model output specifications–including uniform data formats and standard variable names, units, dimensions, etc.

At the same time, the culture of climate change research had evolved to make open data sharing an expected practice. These

two considerations led to the next phase of CMIP, which the WGCM CMIP Panel formulated, and PCMDI officially agreed to support and host the multi-model data. CMIP3 was formally endorsed by the WGCM members (representing contributing modelling centres) in October 2003.

## 2.3 Coupled Model Intercomparison Project phase 3 (CMIP3)

The considerable growth in coordinated MIPs continued throughout the early 2000s and led to the development of what became

known as the third phase of CMIP, CMIP3 (Meehl et al., 2007a). In the early stages, activities that came to define the project were more identified by discrete subprojects, targeting science questions of ocean realism in comparison to observations (e.g., Orr, 1999; Dutay et al., 2002, 2004), and climate change detection and attribution research targeting the "historical" period (e.g., Hegerl et al., 2003). Subsequently, many of the initial activities evolved into a more coordinated collection of experiments, which entrained an even greater number of contributing modelling groups (Figure 1).

The project saw a step-change across numerous facets, driven by growing and emerging science themes (such as climate change commitment) and an increasing appetite from a growing community to access these model data. All contributing models were coupled, simulating the atmosphere, ocean, sea-ice, and in some cases, the first dynamic land components. By this phase, the flux adjustment, used by many CMIP1/2 contributing models, was largely unnecessary for most contributing AOGCM configurations (Durack et al., 2012). There was also a dramatic scientific expansion from the fixed pre-industrial



control (picntrl) and idealized 1 percent compounding $CO_2$ (1pctto2x, 1pctto4x) CMIP2 experiments to include for the first time a "Twentieth-century" simulation ("20th Century Climate in Coupled Models", 20C3M; ∼1860-1999), extending from the mid-19th to late 20th centuries, and directly comparable to the growing observational record (Meehl et al., 2007a, Table A1).

The 20C3M experiment, along with parallel 21st century future projection simulations as documented in the Special Report on Emission Scenarios (SRES 2000-2100; Nakićenović et al., 2000) was a CMIP step-change, for the first time providing

model data that could be directly compared to the growing observational datasets, and outlining a project design that would be replicated to great effect in subsequent phases (to become the DECK and ScenarioMIP in CMIP6). Partial motivation for the 20C3M experiment was to address the growing science around climate change detection and attribution (DandA; e.g., Santer, 1996; Hegerl et al., 2003) a research focus that was rapidly expanding since the "..*discernable* human influence on global climate" statement was published in the IPCC SAR (Santer et al., 1996).

During the planning phase, there was also the explicit recognition that identified unforced model internal variability needed to be better understood, and multi-member ensembles (or runs) were encouraged to enable variability quantification (Meehl et al., 2007a). The project also began to include more experiments, targeting specialized interests proposed by others, expanding CMIP's scientific horizons.

In 2003, for example, the Cloud Feedback MIP (CFMIP) introduced an instantaneous $CO_2$ doubling experiment (2xco2)

and a control (slabcntl), both relying on an ocean with a well-mixed "slab" layer but without representation of the deeper ocean (Manabe and Stouffer, 1979), which was included as part of the official CMIP3 experiment suite (see also Table A1; McAvaney and Le Treut, 2003; Webb et al., 2017).

The concept of committed warming was also addressed systematically for the first time in CMIP3 experiments with stabilized $CO_2$ that were run from the end of the SRES scenarios in 2100 out to 2200, and in some cases to 2300. The idea that even

if $CO_2$ concentrations were stabilized, the system would continue to warm came to be recognized as a central property of the climate system for temperature as well as sea level rise (Meehl et al., 2005b). Additionally, the Paleoclimate MIP phase 2 (PMIP2) was underway - PMIP had been operating in parallel to the A/CMIP phases since 1991 (Villwock and Mitchell, 2003; Braconnot et al., 2011; Joussame and Taylor, 2021).

These were just a few of the internationally coordinated activities of the time. At a combined WGCM and International

Geosphere-Biosphere Programme (IGBP) Global Analysis Interpretation and Modeling (GAIM) meeting in Canada in 2002, the combined leadership recognized community confusion at the rapid MIP proliferation. To better communicate and coordinate activities, a catalogue of MIPs was created on the CLIVAR website, with additions and corrections requested to be sent to the WGCM, GAIM, and CLIVAR Project Office leads. The last update listed more than 30 unique activities, in addition to their AMIP and CMIP precedents (Meehl, 2003) and this list continues to be available on WCRP webpages today.

Like their A/CMIP counterparts, the CMIP3 model output saw even more attention than previous phases. CMIP3 marked the beginning of a historical new phase of climate science research, whereby state-of-the-art climate model output was made widely available to anyone, ranging from students to senior researchers. Thus, CMIP3 started the modern MIP era of open internet access to multi-model climate data. The CMIP3 results, in addition to the body of literature that current and earlier phases of model simulation archives had facilitated, became the foundation of the IPCC Fourth Assessment Report (AR4;



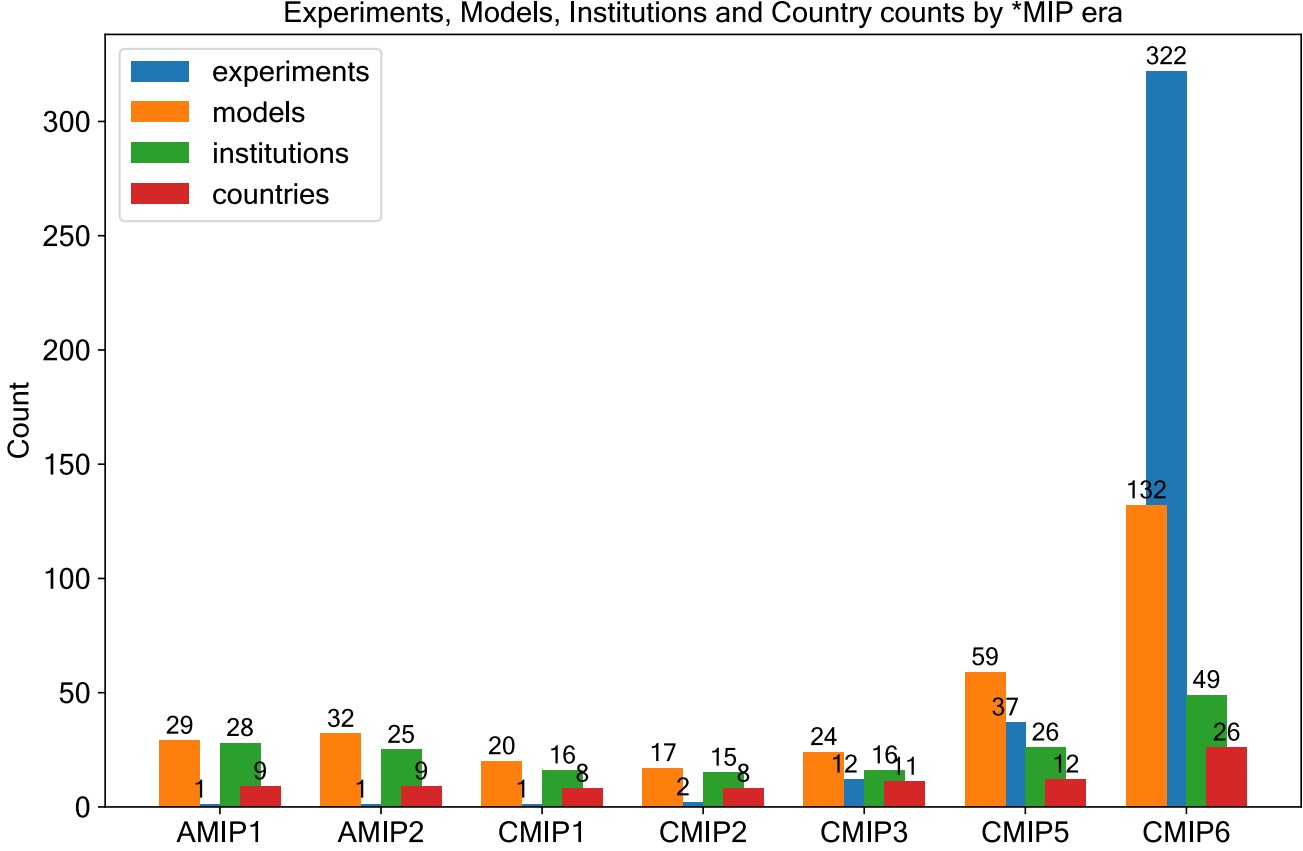

**Figure 1.** Experiments, models, institutions, and countries contributing to completed Model Intercomparison Project (MIP) phases, from 1989 to today. The project's growth over the most recent phase reflects a growing community appetite for the latest climate data to inform decision-making, a distant evolution from the climate science origins of AMIP1. See Table A1 for an expansion of the experiment lists.

Meehl et al., 2007b; Randall et al., 2007), which culminated with the award to the IPCC of the 2007 Nobel Peace Prize (Kerr and Kintisch, 2007).

## 2.4   Coupled Model Intercomparison Project phase 4 (CMIP4)

Though the specified CMIP3 experiments were mostly completed by early 2007, the urgent science questions involved with emerging climate change DandA research subsequently compelled the modelling groups to quickly formulate and run so-called

single-forcing experiments that were particularly in demand.

These single-forcing experiments held all but one of the 20C3M forcings fixed through the 20th century (e.g., WMGHGs-only, sulphate aerosols-only), and were informally undertaken by several modelling groups. Since these were a relatively small set of experiments being run quickly to rapidly provide output to the growing DandA community, the CMIP Panel decided



that a new CMIP phase was not warranted. Therefore, these runs were never formally contributed to a managed multi-model
CMIP archive but were an addendum to the CMIP3 experiment suite (Stouffer et al., 2017). However, these single-forcing
simulations constituted a unique and compelling multi-model dataset that was referred to informally by WGCM membership
as "CMIP4". These single-forcing runs became the precursor to the CMIP6 Detection and Attribution Model Intercomparison
Project (DAMIP; Gillett et al., 2016) that built off early work focused on resolving the regional patterns of greenhouse gas and
sulphate aerosol forcing (Taylor and Penner, 1994; Santer et al., 1995; Hegerl et al., 2000; Gillett et al., 2002; Hegerl et al.,
220 2003).

As planning began for the experiments targeting the next phase of CMIP, whose model simulations would be assessed in the
IPCC Fifth Assessment Report (AR5), it made sense to align the numbering of the next CMIP phase and IPCC report, CMIP5
and AR5 respectively. This additionally also avoided unnecessary confusion with the developing coordinated carbon-cycle
modelling activities, with the Coupled Climate-Carbon Cycle Model Intercomparison Project (C4MIP) an active collaborative
project of the WGCM and the IGBP that had been running in parallel to the prior CMIP phases (e.g., Fung et al., 2000; Cox
et al., 2002; Friedlingstein et al., 2006). C4MIP was one of numerous parallel efforts leveraging the science coordination
nurtured by the growing MIP community.

## 2.5 Coupled Model Intercomparison Project phase 5 (CMIP5)

Late in CMIP3, during the IPCC AR4 process, it became clear that a profound climate change science paradigm shift would
happen. The end of CMIP3 and the publication of the IPCC AR4 in 2007 saw the end of 20 years of relatively coarse grid
AOGCMs, run with climate change non-mitigation scenarios. The new scientific paradigm that was emerging had three el-
ements. First, decadal climate prediction began to use climate models that had been initialized to predict near-term climate
variability over the next decade. Second, Earth System Models (ESMs) with coupled carbon cycle components were being for-
mulated to study longer-term feedbacks past mid-century with new mitigation scenarios. Third, new tangible linkages through-
out the climate science community (WCRP, IGBP, IPCC Working Group II [WG2: Impacts, Adaptation, and Vulnerability],
and IPCC Working Group III [WG3: Mitigation of Climate Change]) were required to advance the science. Consequently, a
landmark Aspen Global Change Institute (AGCI) session, which reflected this new paradigm, was convened in August 2006, to
formulate CMIP5 with participants consisting of climate modellers, chemistry and aerosol modellers, land surface modellers,
biogeochemistry modellers, Integrated Assessment Model (IAM) scientists, and Impacts-Adaptation-Vulnerability (IAV) re-
searchers.

Several "firsts" emerged from the 2006 AGCI session that formulated the CMIP5 experiment design (Hibbard et al., 2007).
CMIP5 marked the first time that the future climate change problem was divided into near-term and long-term timescales,
reflecting a science shift with the emergence of decadal climate prediction and the needs of the stakeholder community for
near-term climate change information. This outcome led to a subsequent 2008 AGCI workshop where the experiment design
for coordinated decadal climate prediction experiments for CMIP5 was formulated, launching a new area of climate science
(Meehl et al., 2009). CMIP5 was the first time ESM experiments were included in a CMIP phase, reflecting the rise of the
carbon cycle being included as standard AOGCM components. The CMIP5 planning process marked the first time the Earth



System Modelling (ESM) Community connected with the Integrated Assessment Modelling (IAM) community to agree on new future emission scenarios (Moss et al., 2008).

Thus, the marked expansion of activities over preceding decades and an ever-expanding data user base led to a considerable project augmentation for CMIP5 (Meehl and Bony, 2011). By this stage, many climate research subcommunities had identified MIPs as an effective way to scientifically collaborate, addressing the dual purposes of answering scientific questions through coordinated experimentation and identifying opportunities for model improvement by identifying common errors and biases across the multi-model ensembles. The developing data standards and delivery infrastructure also considerably aided
collaboration, easing access to obtaining and using CMIP output (see section 4).

For CMIP5, the "core" experiments included CMIP simulations AMIP, piControl, historical (20C3M), 1pctCO2 (previously 1pctto2x, 1pctto4x), and an instantaneous $CO_2$ quadrupling experiment (abrupt4xCO2) included, targeting model behaviour assessment and climate sensitivity, preempting the CMIP6 "DECK". In addition, the success and attention of the three future-focused SRES projections included CMIP3 led to an augmentation and four of the next-generation Representative Concentra-
tion Pathway (RCP) scenarios being identified for inclusion in CMIP5 (Moss et al., 2010).

The importance of the carbon cycle through land and ocean biogeochemistry had also been recognized, which led to several experiments being included forced with $CO_2$ emissions (rather than the imposed atmospheric concentrations of the preceding phases) which depended on ESM configurations to contribute (Hibbard et al., 2007; Meehl and Bony, 2011). In addition, anthropogenic aerosol emissions had also been identified as important climate drivers, and were generated and used by a
number of modelling groups in model configurations including more complex atmospheric chemistry (e.g., Lamarque et al., 2010).

After considerable input and iterations with the international climate science community, the CMIP5 experiment design was approved by the WGCM in 2008 (Taylor et al., 2012b). The CMIP5 project design evolution already reflected the more formal MIP structure defined in the following CMIP6 phase (see section 3, Table A1). Projected future scenarios rcp45 and rcp85 were
defined (Stouffer et al., 2011), with second tier scenarios rcp26 and rcp60 also included (precursors to CMIP6 ScenarioMIP experiments; Hibbard et al., 2011).

CFMIP follow-on experiments included patterned SST experiments sst2030, amip4xCO2, amipFuture, amip4K, along with the aquaplanet experiments aquaControl, aqua4xCO2, and aqua4K (Bony et al., 2011). PMIP follow-on paleoclimate experiments included midHolocene, last glacial maximum (lgm) and past1000 (Braconnot et al., 2011). The project also included ex-
periments targeting short-lived climate forcers sstClimAerosol and sstClimSulfate (Boucher et al., 2011), precursors to CMIP6 AerChemMIP, and the carbon-cycle focused experiments esmrcp85, esmFixClim1, esmFixClim2, esmFdbk1 and esmFdbk2 (Friedlingstein and Jones, 2011), precursors to CMIP6's C4MIP.

The addition of near-term or initialized prediction experiments was also new to CMIP, with these short-running experiments focused on predicting climate states from the next few years through to a couple of decades (Meehl et al., 2009; Doblas-Reyes
et al., 2011). These were identified by sets of initialised hindcast experiments (decadalXXXX and noVolcXXXX) with start years back to at least 1970 (many groups provided coverage back to 1959) to evaluate 10-year prediction skill, as well as near-term future predictions (to at least 2015), precursors to the CMIP6 DCPP activity.





**Table 1.** The Model Intercomparison Projects (MIPs), through time

| Planning begins | 1989 | 1993 | 1995 | 1997 | 2003 | 2008 | 2013 | 2022 |
|---|---|---|---|---|---|---|---|---|
| Project | AMIP1 | AMIP2 | CMIP1 | CMIP2/2+ | CMIP3 | CMIP5 | CMIP6 | CMIP6+ |
| Experiment count | 1 | 1 | 1 | 2 | 12 | 37 | 322 | ∼ |
| "historical" period | 1979-1988 | 1979-2001 | ∼ | ∼ | ∼1860-1999 | 1850-2010 | 1850-2014 | 1850-2022 |
| Data volumes | ∼1 GB$^\dagger$ | ∼500 GB$^\dagger$ | ∼1 GB$^\dagger$ | ∼500 GB$^\dagger$ | 39 TB | 1.5 PB$^\ddagger$ | >16 PB$^\ddagger$ | ∼5 PB |
| Standard output variables/ Tables | 32/4[1] | 114/8[2] | 23/5[3] | 28/5[4] | 143/6[5] | 986/18[6] | 2062/44[7] | ∼ |
| Host infras-tructure | PCMDI FTP | PCMDI FTP | PCMDI FTP | PCMDI FTP | PCMDI FTP; ESG-CET | ESGF, 30 nodes | ESGF, 30 nodes | ESGF |
| Data formats | Fortran formatted binary | Fortran formatted binary, GRIB → P-DRS | GRIB → P-DRS | GRIB → P-DRS | CF netCDF-3 | CF netCDF-4 "classic" | CF netCDF-4 | CF netCDF-4 |
| Data licenses | Registered subprojects | Registered subprojects | Registered subprojects | Registered subprojects | Registered subproject-s/Open | Open | Open, CC-BY 4.0/CC0 | Open, CC-BY 4.0/CC0 |
| Operations begin | 1989 | 1996 | 1996 | 1999 | 2004 | 2011 | 2018 | 2024 |

Notes: sources for "Experiment count" are documented in Appendix A and Table A1. A visual representation of the growth of data volumes is captured in Figure B1.

$^\dagger$Volume estimates are difficult to quantify, as the provided source and the final archived data differ. GRIB and similar precursor formats were reprocessed to netCDF-3.

$^\ddagger$Volume estimates are are for unique project data only. For ESGF federated projects, data was replicated across multiple nodes. Current replicated totals are 5.3 and 27.5 PB for CMIP5 and CMIP6 projects respectively (see subsection 4.10).

Sources for "Standard output variables/Tables" noted by superscripts (1-7) are documented in Appendix B and Table B1.

CC = Creative Commons copyright licenses; CC-BY 4.0 = Creative Commons Attribution 4.0 International; CC0 = Creative Commons Public Domain Dedication; for more information, see https://creativecommons.org/share-your-work/cclicenses/.

CF = NetCDF Climate and Forecast Metadata Conventions (Eaton et al., 2024). P-DRS = PCMDI Data Retrieval and Storage software library, a precursor to NetCDF. FTP = File Transfer Protocol, a common data transfer protocol. GRIB = General Regularly-distributed Information in Binary format, a precursor to NetCDF, which is still used by the weather community today.



## 3 The CMIP6 Project Design

Given the success of the 2006 AGCI session in formulating CMIP5, it was decided to convene an AGCI session in 2013 to
plan CMIP6 and bring together climate scientists, IAM modellers, and IAV researchers recognizing the breadth of science
represented and the broad and growing community of contributors and users these activities were serving. The 2013 CMIP6
planning workshop reflected the rapid advances in climate science three years before the IPCC Sixth Assessment Report (AR6)
schedule was announced in 2016, with a CMIP6 anticipated start date of 2015 extending through 2020. Even though it was
unclear in 2013 whether there would even be an AR6, the CMIP6 schedule reflected the state of the science, and was also
designed to provide advanced planning in case yet another IPCC assessment cycle commenced, to avoid last-minute demands
on the modelling groups to run a lot of experiments over a short period.

As with all previous CMIP phases, an initial exercise was performed to gauge community interest and to assess whether the
modelling groups wanted another CMIP phase (Stouffer et al., 2017). This is because CMIP is a community-driven activity, and
there is no CMIP without contributing modelling groups engagement and support. If the groups did not believe CMIP science
was worth the effort, CMIP would end. The AGCI CMIP6 planning workshop led to a new framing for CMIP in response to
community input whereby it was desired that CMIP6 be a more continuous and distributed activity (Meehl et al., 2014b) with
coordinated experiments for decadal climate prediction experiments formulated at a second AGCI workshop (Meehl et al.,
2014a).

The proposed experiment design led to broad community consultation involving the modelling centres whose simulations
are the tangible substance of CMIP, and the broad and growing communities that rely on CMIP model output for their work
(Eyring et al., 2016a). The formal development of a core suite of experiments was the result, focused on model evaluation,
connecting the activity to the preceding CMIP and AMIP phases. These experiments are referred to as the CMIP DECK (Di-
agnostic, Evaluation, and Characterization of Klima; "klima" being the German word for "climate"). The four primary DECK
experiments, which provided continuity with the experimental protocol of preceding phases, were named amip, 1pctCO2,
abrupt-4xCO2, and piControl. These, along with a historical simulation (1850-2014) and ESM variants of the control and
historical experiments (esm-piControl, esm-hist), were the CMIP6 contributing modelling group entry cards (Eyring et al.,
2016a). The reason for selecting these experiments was that those simulations are typically run by modelling groups during the
model development cycle and, therefore, are not only valuable for benchmarking model performance but also for comparing
different model versions. These were previously considered CMIP5 "core" experiments (Stouffer et al., 2011).
The interest in CMIP6-Endorsed MIPs continued, and the initially endorsed number was 21 (Table 2; Eyring et al., 2016a). It
was recognized that better coordination was needed to align the growing parallel and sometimes overlapping activities with an
awareness of the demands imposed on modelling group resources. In planning CMIP6, the CMIP Panel distributed an open call
for scientifically-focused "community" MIP proposals in April 2014 to encourage and enhance synergies across activities to
address this concern. To reduce duplication and burden on contributing modelling groups, a further standardization around the
common standards and infrastructure developed and delivered in CMIP5 was recommended (Eyring et al., 2016a). The revised



MIP proposals were reviewed in the 2015 Northern Hemisphere summer to prioritize activities that addressed the WCRP Grand Challenges in climate science.

A key feature of the new distributed structure of CMIP6, in response to community and modelling group feedback, was that modelling group participation in CMIP6 was totally elective. The only requirement for participating in CMIP6 was that each

group had to run the DECK experiments. After that, a modelling group could pick from the menu of MIPs and choose to run either none, all, or any number in between. On the WGCM and CMIP Panel side, a MIP could only be approved if a minimum of eight modelling groups committed to running the specified experiments in a given MIP. Once again, this was totally elective on the part of the modelling groups, thus removing the pressure to run and contribute to all experiments like groups had felt obligated to in CMIP5.

Embodying the federated design of MIPs and autonomy, after the Eyring et al. (2016a) CMIP6 overview paper was published, MIP growth and additional experiment registrations were received, augmenting the experiment count from 190 to well over 200 experiments (Balaji et al., 2018). The project growth continued in the following years with two new CMIP6-Endorsed MIPs, the Climate Dioxide Removal (CDRMIP; Keller et al., 2018) and Polar Amplification (PAMIP; Smith et al., 2019) registered in 2018 (see Table 2). During the process of the IPCC AR6 assessment, the development of the COVID pandemic

and some additional science questions led to a further augmentation of CMIP6 official experiments, with the Zero Emissions Commitment MIP (ZECMIP; Jones et al., 2019) and COVIDMIP (Lamboll et al., 2021) being defined, and their experiments were folded into C4MIP and DAMIP, respectively (Table 2). In addition, in early 2020 some sensitivity experiments were also added to the DECK and ScenarioMIP (Durack et al., 2024), enabling a single modelling group to contribute simulations comparing the impact of changed climate forcing from CMIP5 to CMIP6 in a single model large ensemble (e.g., Fyfe et al.,

2021; Holland et al., 2024).

By March 2020, the CMIP6 project included 322 experiments, contributed across 22 MIPs (Table 2). Compared to the prior phases, this order of magnitude growth marked enhanced participation and complexity (Figure 1), reflecting an expansion of climate science and the successful federation of the project structure and management.

CMIP6 also embodied changes that occurred as the climate community itself evolved. In early phases, model simulation and

analysis were carried out within or in collaboration with a small subset of individual groups. Today, modelling groups develop models and routinely release state-of-the-art model output for public scrutiny, with most of the analysis taking place outside the contributing centres. As such, CMIP planning now involves climate modelling groups, the communities that comprise MIPs and their science foci, and the community of scientists and stakeholders analyzing results. These perspectives are routinely included in ongoing consultation and next-phase planning (Stouffer et al., 2017).

While community consultation underpins the CMIP project planning and development to answer pertinent science questions of the day, in the end, as noted above, it is the contributing modelling groups that decide what experiments to prioritize and simulation data to provide for broader community consumption. Subsequently, expanding research and stakeholder communities determine what aspects of available model data will be analysed. The CMIP project's central goal is to advance scientific understanding of the Earth system and its responses to ongoing natural and anthropogenic forcing agents. It aims to provide a



valuable and tangible resource for national and international climate assessments, including the United Nation's IPCC reports and the Global Stocktake (Stouffer et al., 2017).

## 4 CMIP supporting infrastructure and organisations

The early FANGIO success and the developing appetite to improve climate change understanding led to the establishment by the US DoE of the Program for Climate Model Diagnosis and Intercomparison (PCMDI) at Lawrence Livermore National

Laboratory (LLNL) in 1989. PCMDI was created to develop improved methods and tools for the diagnosis, validation, and intercomparison of global climate models. LLNL was selected to host the PCMDI to leverage the co-location of the US DoE Research Scientific Computing Center (NERSC) computing resource (1974-1996) and the pre-existing DoE-supported atmospheric climate modelling efforts that were already underway (e.g., MacCracken and Luther, 1985; Potter et al., 2011).

The existence of the NERSC high-performance computing infrastructure delivered a significant opportunity to the program,

enticing 12 international modelling groups to participate in AMIP1 and providing compute cycles to run their codes to develop simulations. These groups ran the ten simulation years (1979-1988), storing their output on the same system (Gates, 1991, 1992a). By 2000, there was a recognition that supporting infrastructure, data standards, software, and hardware development was essential for project success. The infrastructure facilitated data delivery and was becoming heavily relied upon to serve the growing MIP-contributing communities and downstream users. As such, "MIP" infrastructure was now of equal

importance to the parallel intercomparison, science coordination, and experimental protocol activities that it served, with the "I" in the MIP acronym interchangeable across *intercomparison* and *infrastructure* terms (Gleckler, 2001).

Before CMIP5, PCMDI and its collaborators managed all the data distribution responsibilities for AMIP and CMIP (see subsection 4.4). Early in CMIP5 planning it became clear that PCMDI could not manage the expected number of data producers and consumers alone. PCMDI requested help from the Global Organisation of Earth System Science Portals (GO-ESSP;

see Williams et al., 2009), to help design and deliver a globally distributed data infrastructure. The resulting internationally-federated infrastructure became the Earth System Grid Federation (ESGF), with the name itself an evolution, like the software, of the preceding DoE "the Earth System Grid" initiative (see subsection 4.4; Williams et al., 2009).

With scale and the advent of the ESGF, the entire CMIP process began to depend more on the underpinning technical infrastructure (which, as is covered below), expanded well beyond the ESGF data publication and dissemination alone. To

coordinate infrastructure development and delivery, the WGCM established the WGCM Infrastructure Panel (WIP) in 2014, working in parallel and in collaboration with the CMIP Panel, facilitating the production and consumption of CMIP data products. The WIP maintains the necessary standards and policies for model data sharing and ensures that the various infrastructure components are integrated and function smoothly. Its work was critical to the success of CMIP6.

For CMIP5, and since, many independently-funded efforts contributed to the supporting infrastructure. Amongst them, key

roles were played by the Climate and Forecast (CF) Metadata Conventions (subsection 4.1), used to standardize the content of model output files, and the CMIP Controlled Vocabularies (CVs; subsection 4.2), which defined a limited set of terms used to describe and uniquely identify datasets and to aid in searching for data of interest. These standards are now integral to



**Table 2.** The CMIP6-Endorsed Model Intercomparison Projects (MIPs), their science focus, and citation

| Count | MIP identity | MIP Description | Science focus | Citation |
|---|---|---|---|---|
| 1 | AerChemMIP | Atmospheric Aerosols and Chemistry | Short-lived climate forcers | Collins et al. (2017) |
| 2 | C4MIP | Climate-Carbon Cycle | Carbon cycle | Jones et al. (2016) |
| 3 | CDRMIP | Carbon Dioxide Removal | Carbon dioxide removal | Keller et al. (2018) |
| 4 | CFMIP | Cloud Feedback | Cloud feedbacks | Webb et al. (2017) |
| 5 | CMIP | Diagnostic, Evaluation and Characterisation of Klima (DECK) | Core evaluation, link model evolution across A/CMIP eras | Eyring et al. (2016a) |
| 6 | DAMIP | Detection and Attribution | Climate change detection and attribution | Gillett et al. (2016) |
| 7 | DCPP | Decadal Climate Prediction Project | Initialized climate prediction | Boer et al. (2016) |
| 8 | FAFMIP | Flux-Anomaly Forced | Idealised forced ocean responses | Gregory et al. (2016) |
| 9 | GeoMIP | Geoengineering | Geoengineering intervention | Kravitz et al. (2015) |
| 10 | GMMIP | Global Monsoons | Monsoon climatology, variability, prediction and projection | Zhou et al. (2016) |
| 11 | HighResMIP | High Resolution | High resolution (<50 km) | Haarsma et al. (2016) |
| 12 | ISMIP6 | Ice Sheet | Ice sheet (Greenland & Antarctica) | Nowicki et al. (2016) |
| 13 | LS3MIP | Land Surface, Snow and Soil moisture | Land-only simulations | Van Den Hurk et al. (2016) |
| 14 | LUMIP | Land Use | Land-use and land-cover change | Lawrence et al. (2016) |
| 15 | OMIP | Ocean | Ocean-only simulations (physics) | Griffies et al. (2016) |
| - | OMIP | Ocean | Ocean-only simulations (biogeochemistry) | Orr et al. (2017) |
| 16 | PAMIP | Polar Amplification | SST, sea-ice roles in polar warming | Smith et al. (2019) |
| 17 | PMIP | Paleoclimate | Climate of the deep past | Kageyama et al. (2018) |
| 18 | RFMIP | Radiative Forcing | Radiative forcing, transfers and model responses | Pincus et al. (2016) |
| 19 | ScenarioMIP | Scenarios | Future climate change projections | O'Neill et al. (2016) |
| 20 | VolMIP | Volcanic forcing | Volcanic forcing climate responses | Zanchettin et al. (2016) |

**Table 2 continued overpage..**

Notes: MIPs 1 through 20 above are registered in the CMIP6 Controlled Vocabulary (CMIP6_CVs) GitHub repository, the primary source of CMIP6 registered information. Due to inflexible infrastructure, experiments defined by MIPs 21 and 22 were registered under an existing MIP (C4MIP and DAMIP, respectively).





**Table 2.** The CMIP6-Endorsed Model Intercomparison Projects (MIPs), their science focus, and citation (continued)

**MIPs (added as an existing MIP subactivity)**

| | | | | |
|---|---|---|---|---|
| 21 | C4MIP: ZECMIP | Zero Emissions Commitment | $CO_2$ emissions and remaining carbon budget | Jones et al. (2019) |
| 22 | DAMIP: COVIDMIP | Modifying emissions to account for COVID-19 | Updated GHG, aerosols, ozone, and optical properties in response to COVID-19 lockdowns | Lamboll et al. (2021) |

**Diagnostic MIPs (requesting output from CMIP6 experiments, but having no MIP child experiments, and no data published to the ESGF CMIP6 project)**

| | | | | |
|---|---|---|---|---|
| | WCRP CORDEX | WCRP COordinated Regional Downscaling EXperiment | Regional downscaling related variable request | Gutowski Jr. et al. (2016) |
| | DynVarMIP | Dynamics and Variability | Momentum and energy transport and model bias related variable request | Gerber and Manzini (2016) |
| | SIMIP | Sea-Ice MIP | Sea-ice related variable request | Notz et al. (2016) |
| | VIACSAB | Vulnerability, Impacts, Adaptation and Climate Services Advisory Board | Climate change impacts related variable request | Ruane et al. (2016) |
| **Count** | **MIP identity** | **MIP Description** | **Science focus** | **Citation** |

Notes: MIPs 1 through 20 above are registered in the CMIP6 Controlled Vocabulary (CMIP6_CVs) GitHub repository, the primary source of CMIP6 registered information. Due to inflexible infrastructure, experiments defined by MIPs 21 and 22 were registered under an existing MIP (C4MIP and DAMIP, respectively).

interpreting and accessing the growing model data archives. PCMDI developed software (the Climate Model Output Rewriter; CMOR) to impose the use of these conventions and vocabularies in data production, and the process of putting data into the
right format with the correct information became known as "CMORization" whether or not CMOR was used.

Data conforming to the necessary standards can be published into the ESGF, which is findable and accessible worldwide. A subset of the CMIP archive is also replicated at the IPCC Data Distribution Centre (IPCC DDC; subsection 4.5), where augmented with extra metadata, it is expected to be preserved and made available indefinitely.

Additional initiatives to collect and organize model and experiment documentation (see subsection 4.7), to establish a data
citation service (issuing DOIs for published datasets; subsection 4.8), and to make errata information about the data (subsection 4.11) all also fell under the purview of the WIP.

## 4.1 Climate and Forecast (CF) Metadata conventions

By the early 1990s, interest in sharing the output of climate models with researchers outside the originating groups gained momentum. This led to greater attention to clearly defining the model output quantities, ensuring that datasets were self-
describing to avoid misinterpretation, standardising their digital formats to ease data transfers and use, and documenting the model configurations that produced the data. In parallel, there were computer science developments focused on generating



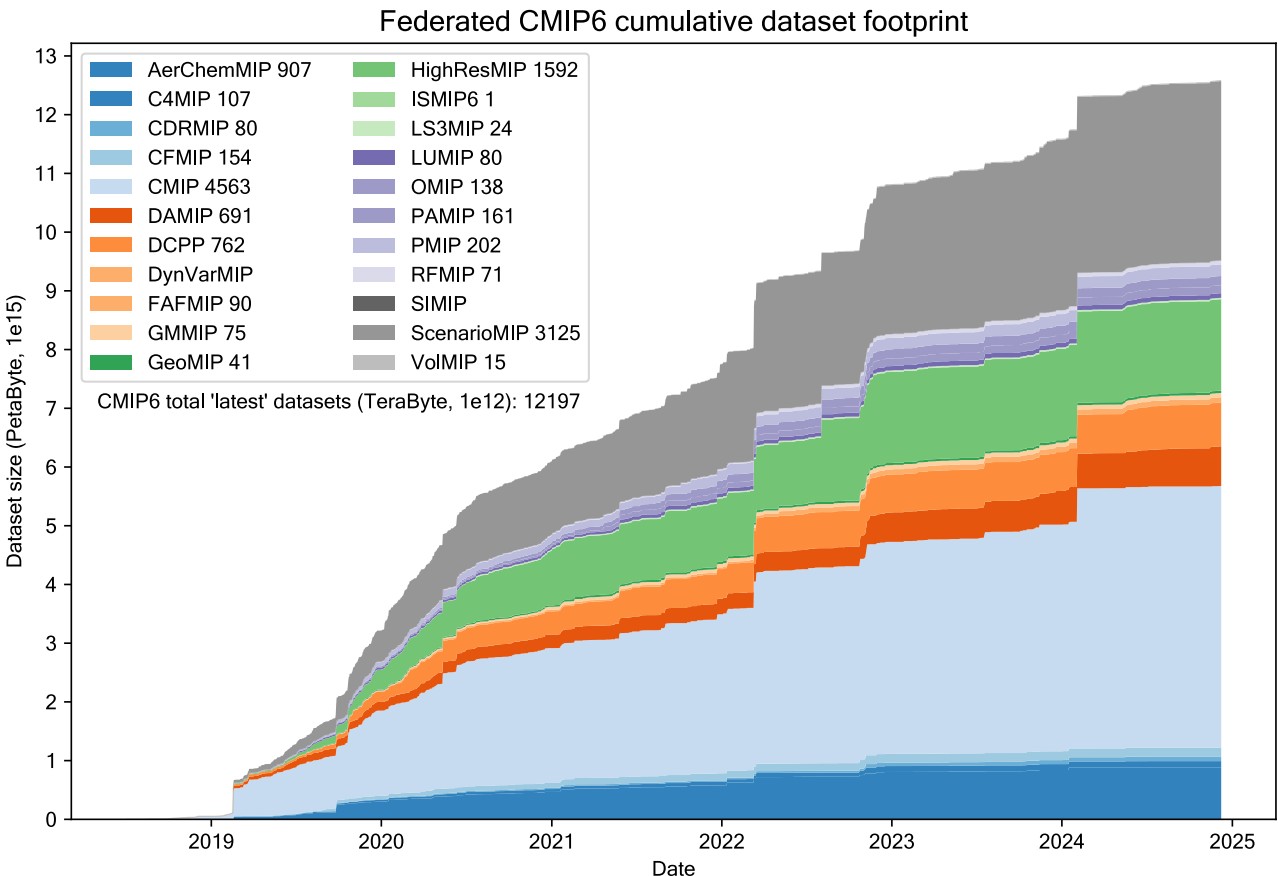

**Figure 2.** Growth of the CMIP6 project over time, with data sizes (PB) identified by colours for each of the CMIP6 endorsed MIPs (see Table 2) from the initial date of data publication (9th May 2018) through to the latest publication (9 December 2024). The legend denotes MIP project sizes in Terabytes (TB). The largest data contributions in CMIP6 are from the CMIP (light blue) and ScenarioMIP (dark grey) activities, comprising 8.4 PB of the >15 PB total. Data growth fluctuated throughout the project's life, with current 2024 publication rates of 2.1 PB/yr (prorated). This compares to previous years: 0.8 in 2023, 3.3 in 2022, 1.8 in 2021, 3.1 in 2020, 3.3 in 2019, and 0.1 PB in 2018.





digital formats that met the needs of growing meteorological data streams, with an aspiration to develop portable and self-describing file formats (where data and metadata that described it were co-contained in files). These developments led to numerous early releases of the network Common Data Format (netCDF) through the late 1980s and early 1990s. In May 1997,

netCDF 3.3 was released, which included a new type-safe interface for the C and Fortran languages. In March 2001, netCDF 3.5.0 was released, which integrated a new Fortran90 interface, making this library suitable for integration with climate model libraries primarily written in Fortran (University Corporation of Atmospheric Research, 2021). netCDF was explicitly designed to meet the goal of self-describing data, to be used in conjunction with community-developed conventions in development, and to facilitate progress; the Unidata Program Center (Unidata, US) hosted a directory of these evolving efforts.

In parallel to the netCDF digital format and language interface development, the ability to store rich, self-describing metadata within the file led to considerable work on data conventions. Of climate data relevance, the first was developed by the Cooperative Ocean/Atmosphere Research Data Service (COARDS), a US National Oceanic and Atmospheric Administration (NOAA) and US university cooperative aimed at sharing and distributing global atmospheric and oceanographic research datasets. COARDS 1.0 was released in February 1995 and laid out data variable conventions such as assigning spatial di-

mension/coordinate attributes (height, latitude, longitude) and data variable attributes (long_name, missing_value, units), in addition to some required global attributes (e.g., title) providing descriptive context for the data. Much of the COARDS work aided the Ferret visualization and analysis environment development, with Ferret 4.0 released in March 1995 (NOAA PMEL, 1995). This data standardisation aided the coordination of software efforts, reducing parallel development into a couple of well-supported packages targeting the evolving COARDS conventions.

By 1997, the large and growing MIP archives had accumulated tens of thousands of output files from dozens of model configurations. The diverse and growing international community interested in analyzing and applying results from the models was hampered by the diversity of data formats, structures, and nomenclatures that climate modelling centres had adopted. There would be obvious benefits in standardizing the data. To fill the void, the Gregory, Drach, and Tett (GDT v1.4; Gregory et al., 1999) standards were first introduced in June 1997. The GDT standard expanded markedly on COARDS, focusing on better

enabling GCM output to be self-describing and to conform to standard conventions. These conventions enabled the explicit definition of the location and sizes of the grid cells, coordinates specified by tuples [CF "formula_terms"], multiple coordinate systems for a given data variable [CF "grid_mapping"], provisions for time series and vertical profiles at points and trajectories [CF "discrete sampling geometries"], and metadata describing subgrid variation [CF "cell_methods"].

In parallel with the development of GDT, work had been going on the National Center for Atmospheric Research (NCAR,

US) on the Community Climate System Model (CCSM) netCDF Conventions (CCSM netCDF Conventions). Both conventions were superseded following a meeting in December 1999 at NCAR, when it was decided to merge these efforts, leading to a collaboration of a larger group of authors who began work developing what became known as the NetCDF Climate and Forecast (CF) Metadata Conventions. Building on and going far beyond COARDS, GDT, CCSM netCDF Conventions, and similar efforts, version 1.0 of the CF Conventions was released in October 2003 (e.g., Eaton et al., 2024).

Subsequently, the CF conventions were adopted by CMIP, starting in phase 3. Since that time, CF gained in popularity across climate sciences and beyond. By 2006, the contributor community was rapidly growing, and a formal governance structure




was established in response (Lawrence et al., 2006). Today, an ever-expanding group of volunteers maintains and continues to augment these conventions (see https://cfconventions.org). The evolution (while maintaining backward compatibility) has been ongoing through several releases, the latest being version 1.12-draft. The stability of CF, the vibrant community supporting it,
and the development of software that can interpret CF data have been foundational to the widespread use of CMIP data.

From the inception of CF, in parallel with the elaboration of the conventions, work has proceeded on compiling the CF standard name table. Originating in the "quantity" concept of the preceding GDT conventions (Gregory et al., 1999), a standard name is a self-explanatory phrase identifying a particular geophysical quantity. CF currently defines about 4900 standard names. Standard names have become an element of CMIP controlled vocabularies, work which began with AMIP1, and were
formalised during CMIP3 (see subsection 4.2).

From the inception of CF, parallel work continued compiling the CF standard name table. This idea originated as a "quantity" concept in the preceding GDT conventions (Gregory et al., 1999). This ongoing work led naturally to the development of CMIP Controlled vocabularies, which were formalised during CMIP3 (see subsection 4.2).

## 4.2  Controlled Vocabularies (CVs)

As coordinated international climate science activities grew from AMIP1 and 2 through the early CMIP phases, it was recognized that imposing increasingly detailed requirements on the data reported would facilitate its use and increase its value. Early on, it was decided that rather than store multiple variables in a single file, the model output should be organized such that each file contained a single variable. This was done to reduce data download volumes since most analysts of the CMIP data were expected to consider only a small number of variables.

Beginning with CMIP3, each requested variable was assigned a unique name (e.g., "ts" for surface temperature and "pr" for precipitation rate), and all data were written compliant with the CF conventions (e.g., Taylor et al., 2009). Additionally, the names of the coordinates and their ordering were standardized, and certain CF optional attributes were made mandatory (e.g., "cell_methods"; Eaton et al., 2024). Standardisation simplified analysis codes, which could now ingest data produced by multiple models in a common way. In addition, specific metadata were required to be recorded in each file identifying the
source of the data (institution, model name, version) and experiment conditions (e.g., experiment name). Thus, the detailed specifications for CMIP3 (see Taylor and CMIP Community, 2005) imposed uniformity on the format and structure of the data and metadata that facilitated its use.

The first collection of controlled vocabularies (CVs) was also introduced in CMIP3 (e.g., Taylor and CMIP Community, 2005). In its simplest form, a CV lists the terms used to describe a specific dataset attribute. In CMIP3, there were CVs for
variable names (e.g., "pr", "tas", "ua", this subsequently evolved into "standard output" see subsection 4.3), experiments (e.g., "amip", "picntrl", "SRESB1"), institutions and models (e.g., "BCC-CM1", "GFDL-CM2.1", "UKMO-HadGEM1"), and table names (e.g., "O1", "A1", "A2", "A3", see Table B1). The table names identified collections of variables reported at a common "frequency" (1 = monthly, 2 = daily, 3 = 3-hourly) and were all associated with either the atmosphere (A) or ocean (O). The CVs were used to construct the directory structure for the data archive, with output from different experiments, models, and
variables placed in separate sub-directories. The structure ensured that although the loosely defined template for naming files



would not ensure name uniqueness across the entire CMIP3 archive, it would ensure that the names were unique within each lowest-level sub-directory.

The increased uniformity of the CMIP3 data requirements facilitated subsequent research. Still, it burdened modelling groups, who were required to rewrite their data in a new way and a format unfamiliar to some. To assist groups in com-
plying with the model output requirements, the Climate Model Output Rewriter software (CMOR1; Taylor et al., 2006) along with the CMIP3-CMOR-Tables (Doutriaux and Taylor, 2005) were developed and made available. In the tables, the 143 requested variables (see Table 1) were grouped so variables likely to be processed under a common procedure appeared together (e.g., ocean and atmosphere variables were found in separate tables). The detailed data specifications and the clearly defined variables were documented in the CMIP3-CMOR-Tables and an "output requirements" document (Taylor and CMIP Commu-
nity, 2005). Once the modelling groups prepared their output according to the specifications, it was collected and organized on the PCMDI computing system in an easily searchable directory structure. The data could be searched and downloaded directly via anonymous FTP or the ESG web-based data service.

In CMIP5, the trend toward further standardization continued as the project expanded to include additional specialized MIPs, 26 modelling groups, and 37 experiments (Table 1, Figure 1). The original CMOR1 FORTRAN software was revised and recast
as C code with FORTRAN and Python interfaces (Doutriaux and Taylor, 2011). New CMIP5-CMOR-Tables (Doutriaux and Taylor, 2013) were defined to accommodate the dramatic increase in the variables requested, which now totalled 986 (see Table 1). The data requirements were inherited mainly from CMIP3 but were augmented with additional required attributes, some with values drawn from CVs (e.g., "frequency" and "modelling_realm").

A major advance in enabling communication among the various components of software infrastructure supporting CMIP5
was the development of a "Data Reference Syntax" (DRS; Taylor et al., 2012a). The DRS relied on several dataset descriptors and associated CVs that uniquely defined each dataset contributing to the CMIP5 archive. For example, the DRS elements were used to define the CMIP5 directory structure and to construct filenames that were guaranteed to be unique across the entire archive. The DRS was also essential for communication and sharing among the data nodes, which comprised a now federated ESGF data distribution infrastructure (subsection 4.4; Williams et al., 2011). The DRS dataset descriptors were supplemented
with dataset version information assigned during the data preparation and ESGF publication processes, which greatly aided the replication of datasets across the major federated ESGF data nodes.

For CMIP6, the evolution of the data standards, CVs, and supporting infrastructure continued as the project ballooned in size to include 22 specialized "endorsed MIPs" (now referred to as "Community MIPs"), 49 contributing modelling groups, and 322 experiments (Table 2, Table 1, Figure 1). Due to the broadening of model diversity and configurations and the concomitant
increase in variables of interest, a new approach to defining the data request was tried whereby the needed variables were placed in a database that could be accessed via an API or displayed in different human-readable forms (see subsection 4.3). In total, 2062 variables were requested across all experiments, and these were all transferred into the CMIP6-CMOR-Tables (Nadeau et al., 2017) and were relied on by a revised CMOR3 (Mauzey et al., 2024) (also see Table D1).

As new services that provided centralized access to extensive model documentation (see subsection 4.7) and enabled formal
data citation (see subsection 4.8) began supplementing CMIP's basic service of providing access to CMIP data through ESGF





and the IPCC DDC (see subsection 4.4 and subsection 4.5), the CVs became more critical for maintaining consistency across the infrastructure components. The earlier practice of publishing simple human-readable lists of descriptors to identify CMIP datasets had resulted in different parts of the infrastructure replicating this CV information in different specialized formats more suitable for specific purposes. In some cases, the information needed to populate a CV was gathered independently by

different components of the infrastructure (e.g., the acronyms used to identify each CMIP experiment), which led to some inconsistency in CV content that hampered transparency and communication across the infrastructure and sometimes confused users of the CMIP data. Consequently, the WIP decided that for CMIP6, a reference repository of CVs would be established, and all infrastructure components would rely on this reference source (Durack et al., 2024). The CVs were in JavaScript Object Notation (JSON) files, which humans and machines read easily. Building on the CVs relied on in CMIP5, a few new ones

were added. Still, the main advance was providing a definitive source for defining the terms used to uniquely identify CMIP6 datasets and managing them transparently on GitHub.

### 4.3 Variable request and standard output

For any simulation, models produce fields of climate- and weather-relevant information covering temporal scales from minutes to centuries. Not all information of conceivable interest can be archived due to resource constraints regarding post-processing

effort and available storage. Experiment planners must anticipate how essential each model-produced field will be for achieving its objectives and decide whether the fields should be reported at annual, monthly, daily, or sub-daily time-intervals. The challenge is to limit the "standard output" requested from a model experiment to the minimum required and then request, at a lower priority, other variables of interest that are not critical to achieving the primary objectives.

In the pre-MIP era (see Table 1), the FANGIO experiments were focused almost entirely on a single objective: to determine

whether clouds were responsible for the range of global climate sensitivities found in models. The narrow objective meant that the required output was limited to a few fields and, primarily, to their global means.

When AMIP was proposed, the potential scope of objectives expanded considerably, spanning both time-scales and spatial scales, driven by a growing contributor community. The most popular and readily analysed climate variables were monthly means. Still, even in the early days of AMIP1, scientists were interested in process-level analyses to understand model re-

sponses, which required daily or even higher temporal frequency fields.

In AMIP1, storage and resource limitations were a primary concern in defining the "standard output" list. The requested output was agreed upon following extended discussion and negotiation involving the WCRP WGNE, PCMDI, contributing modellers, and other keenly-engaged scientists. Enough temporal mean data was saved to serve the numerous subprojects that were subsequently established to systematically evaluate a diversity of model characteristics (Gates, 1995).

Seeing the interest in AMIP grow, modelling groups realized it would be beneficial to expand the set of model outputs requested to enable a broader range of analyses. Thus, in AMIP2, the standard output more than tripled (from 32 to 114; Table 1) and for the first time included 6-hourly data, which was particularly useful in studying weather phenomena. In addition, the monthly mean covariances needed to evaluate a model's Lorenz energy cycle were reported. These covariances enabled the intended analysis (Boer and Lambert, 2008), but modelling groups felt that given their limited use, the effort required in





computing them was unwarranted. Since AMIP2, covariances have not been part of the routine output request and have only
been saved for targeted studies.

Compared with AMIP2, CMIP1's standard output list was modest, with just 23 variables requested (Table 1). The sole
CMIP1 experiment was a control run focusing on mean climatology, and most variables requested were summer and winter
mean fields. There were, however, three multi-year monthly mean surface fields requested: surface air temperature, precipita-
tion, and mean sea level pressure. As in AMIP, the scientists directly involved in organizing CMIP1, primarily the members
of the SGGCM/WGCM (see subsection 2.2) and PCMDI, decided which variables would be archived. Also, as in AMIP, the
initial standard output list would inevitably grow over subsequent phases as its scientific impact became apparent.

Although CMIP2 saw only a tiny increase in CMIP's scope, in CMIP3, the number of experiments and the standard output
requested increased by more than a factor of five. CMIP3 output was expected to serve an increasingly wide diversity of
scientific analyses, and the historical and future scenario experiments would begin serving those studying climate impacts. For
such purposes, the variables of most interest to those developing models were augmented with variables characterizing changes
near the surface and climate "extreme indices" that might be used to assess impacts. Like the AMIP2 covariance statistics, the
requested extreme indices proved challenging to compute and were eliminated from subsequent standard output lists. This set a
good precedent for subsequent phases. Still, culling variables from the standard output based on usage statistics should perhaps
be pursued more vigorously in future phases even though data popularity is only one of several criteria that must be considered.

As in earlier phases, the CMIP Panel and PCMDI coordinated the effort to define the CMIP3 output, but input was sought
from those with specialized scientific expertise in areas such as clouds (e.g., regarding model variables required for the In-
ternational Satellite Cloud Climatology Project [ISCCP] simulator needed by those involved in the Cloud Feedbacks Model
Intercomparison Project [CFMIP]) or impacts (e.g., regarding extreme indices).

CMIP5 represented a second step change in the size and complexity of the standard output request. The number of experi-
ments tripled to 37, and the number of different variables increased by a factor of 6 to 986 (Table 1). For the first time, CMIP
attempted to coordinate the experiments designed by multiple, independently managed MIP communities (and all experiments
were briefly described by the CMIP5 overview paper, Taylor et al., 2012b, also see Table A1). As in CMIP3, a common set
of standards was imposed on model output. A single comprehensive list of variables was compiled and organized into 6 tables
and 13 sub-tables (Table 1, Table B1; Taylor and CMIP Community, 2013).

For each MIP experiment defined, whole groups of variables were requested by specifying a subset of the defined tables; all
the variables in a requested table were expected to be reported for specified portions of each simulation (most commonly the
entire simulation). Each table contained variables that, with rare exceptions, were all produced by a single model component
(e.g., atmosphere, ocean, sea ice, or land) and reported at a single frequency (e.g., 3-hourly, daily, monthly). CFMIP had an
interest in many specialized variables from a small subset of the MIP experiments, and five special tables were defined to meet
CFMIP's unique needs (divided into ten sub-tables; Table B1). All CMIP5 variable tables were accessible as spreadsheets or
as machine-readable text files. The mapping of experiments to the tables/variables that were requested could only be done by
a person reading the spreadsheet, which poorly served automated data preparation job streams.



For CMIP6, fewer limits were imposed on what experiments and which variables requested would be accommodated. Models were also increasingly becoming more comprehensive, evolving, and improving with fewer physical parameterisations and incorporating atmospheric chemistry and biological components necessary to quantify carbon cycle changes. With more experiments and scientists with varied interests involved and using the downstream data, the requested output became transformed from being defined by a limited but dedicated group of experts guided by the CMIP Panel and PCMDI to one involving more extensive community consultation led by the UK Centre for Environmental Data Analysis (CEDA) and taking input from the contributing Endorsed MIP Co-chairs. The CMIP Panel and those preparing the standard output list sought engagement with those studying the impacts of climate change and planning adaptation strategies. That broad community was represented by a Vulnerability, Impacts, Adaptation and Climate Services Advisory Board (VIACS AB; Ruane et al., 2016), which requested that some 477 variables be saved from specific CMIP6 experiments. The standard output list and the mapping of subsets of that list to particular CMIP6 experiments became known as the "Data Request" (Juckes et al., 2020). The rapid and accelerating growth of MIPs (Table 2) and experiments defined across the phases doubled the total variables requested to 2062 (Table 1).

The ambitious CMIP6 data request was largely successful because the number of variables contributed by at least three modelling groups more than doubled from about 670 in CMIP5 to 1500. Still, it is disappointing that two or fewer groups provided some 562 CMIP6 requested variables, and no one provided 185. Although in CMIP5, there were also many under-reported variables, the fraction of these rose in CMIP6: variables reported by no model increased from 4% in CMIP5 to 9% in CMIP6, and the fraction reported by fewer than two groups increased from 22% to 27%. A partial explanation for this may be found in the tight timelines imposed on CMIP6 and the complexity of the task of the data request, which was coordinating requested contributions from models that themselves were becoming more comprehensive in replicating Earth processes. This meant that no single individual from a modelling centre or anyone of the experts coordinating CMIP infrastructure could provide a careful, comprehensive review of the data request. Consequently, several variables that were difficult to produce or thought by modelling groups to be of insufficient interest to warrant inclusion, cannot be found in the data archive. Given the CMIP5 and CMIP6 record of under-reported variables and given that the value of an intercomparison involving results from two or fewer modelling centres is limited, it is suggested that future data requests devote more effort determining which variables are likely to be under-reported so that those variables can be eliminated from the request. This would avoid the extra efforts undertaken by exceptional groups to honour the request. More generally, a clear communication of the importance of each variable and why it is being requested would, perhaps, animate modelling groups to contribute a larger fraction of the requested variables.

## 4.4 The Earth System Grid Federation (ESGF)

While addressing current climate science questions through coordinated experimentation has been CMIP's core focus, facilitating data access and use by a rapidly expanding international community has also been a central project priority. As data standards and controlled vocabularies were being defined and with standard output being requested, a system was developed to host, manage, and make available the data.





For AMIP1/2 and early CMIP phases, the requested standard output was shipped from modelling centres on hard drives or, in some cases, transmitted via FTP to PCMDI. To facilitate management and analysis, PCMDI developed data formats and software packages for storing and visualizing gridded climate data. Rather than attempt to read data stored in several different native formats and structures, the contributed data was rewritten into the PCMDI "standard" DRS (Data Retrieval and Storage; Drach and Mobley, 1995) format, a netCDF file format precursor, and these data were made available to AMIP subproject registrants through a PCMDI file transfer protocol (FTP) server (see Table 1; Gates, 1995).

By 2000, there was recognition in the US that connecting parallel climate science activities and the associated data archives across institutes could maximize investments across independently funded projects. Besides the PCMDI single-site archive of internationally-contributed AMIP and CMIP data, other institutes were beginning to share data proactively. Activities of this kind were underway at the National Center for Atmospheric Research (NCAR, US) with the Community Climate System Model (CCSM) and the Carbon-Land Model Intercomparison Project (C-LAMP) hosted at Oak Ridge National Laboratory (ORNL, US), and copies of data produced by several projects were hosted on the NERSC system at the Lawrence Berkeley National Laboratory (LBNL, US; 1996-). These groups began collaborating to develop a federated US infrastructure supporting data sharing and archival. This became known as the "Earth System Grid" (ESG I; Bernholdt et al., 2007) and was supported by DoE. It quickly gathered momentum with an extension of support from the DoE Scientific Discovery through Advanced Computing (SciDAC) program in 2002 in a second phase (ESG II; Williams et al., 2009). By late 2004, ESG II began distributing the CMIP3 data in preparation for the IPCC Fourth Assessment Report (AR4). By the conclusion of CMIP3 in 2009, the 39 TB archive of model data (see Table 1) had accumulated an order of magnitude more downloads (470 TB; Ananthakrishnan et al., 2007; Williams et al., 2009). Throughout the project's lifetime, the hardware hosting the CMIP3 archive had occasional failures, and there was a recognition that the centralized dependence on a single operational system was unsustainable in the long term, particularly with the growing demand to access and download these data.

By 2006, planning for the IPCC Fifth Assessment Report (AR5) was underway, and with it, CMIP5 (subsection 2.5). The previous ESG successes led to the development of the next-generation Earth System Grid Center for Enabling Technologies (ESG-CET), which involved an expanded collaboration of institutions including Argonne National Laboratory (ANL, US), Los Alamos National Laboratory (LANL, US), National Oceanographic and Atmospheric Administration Pacific Marine Environmental Laboratory (NOAA-PMEL, US) and the University of Southern California Information Sciences Institute (US) (Ananthakrishnan et al., 2007), supported by the US DoE Offices of Advanced Scientific Computing Research (ASCR) and Biological and Environmental Research (OBER).

By 2008, infrastructure planning for CMIP5 had commenced, and the need for a global data infrastructure was clear because PCMDI could not manage the volume of data and community of users, even with ESG-CET support. Recognition was building that there might be opportunities to connect the somewhat US-centric ESG-CET with complementary efforts elsewhere (e.g., Williams, 2011; Williams et al., 2009, 2011). An agreement was signed in December 2008 between PCMDI, the British Atmospheric Data Centre (BADC and, subsequently, CEDA, UK), and the World Data Centre for Climate, German Climate Computing Centre (WDCC DKRZ, Germany), committing to development and support of the international Earth System Grid Federation (ESGF), which would, as a primary responsibility, serve the needs of CMIP and related activities. The two European





centres were leads of the IPCC Data Distribution Centre (IPCC DDC; subsection 4.5), collating and archiving the climate model data to prepare previous IPCC Assessment Reports. The ESGF goal was to develop a distributed federated architecture with dedicated storage resources at each of the three centres to facilitate the publication of CMIP5 model data at the most convenient location. This data could then be replicated across the federation, allowing the users to download or access CMIP5 data from whatever ESGF node served them best (Williams et al., 2011). In addition to the three core centres, additional ESGF nodes were established at the National Computational Infrastructure (NCI, Australia), the National Aeronautics and Space Administration Jet Propulsion Laboratory (NASA-JPL, US), NERSC (LBNL, US), ORNL, and NCAR. During peak interest in CMIP5, the ESGF had 30 active nodes publishing data from contributing modelling groups (see Table 1; Williams et al., 2016).

In 2013, as CMIP6 planning commenced, the international ESGF collaboration leaders became concerned that it would be difficult to meet the expectations placed on it to deliver a federated robust operational system for supporting an archive of CMIP data that would serve without interruption many hundreds of users and perhaps several thousands. To scope how the expectations might be met, the WIP prepared a series of white papers describing not only the requirements for ESGF but for the many related services (including documentation and citation) that would interact with it (Balaji et al., 2018). The challenge facing ESGF developers was that there were no expectations that funding to support an operational system would materialize. Despite this, the ESGF leadership together with the WIP formed a CMIP Data Node Operations Team (CDNOT), which would provide a means of testing the next generation ESGF software and facilitate communication between the software development team and those who were implementing the software at sites (nodes) around the world (Petrie et al., 2021).

In the first half of 2018, the CDNOT organized a series of five "data challenges", designed to ensure that all the ESGF sites participating in the distribution of CMIP6 data would be ready to accept CMIP6 data. As the data challenge activity advanced through its phases, an increasingly complex software ecosystem was tested, and the data volumes the system was required to handle expanded. Such a systematic approach to preparing the quasi-operational software support for CMIP was a major advance. It enabled ESGF to successfully test and install its software at sites maintained by those not directly involved in its development. This ensured a smoother provision of services once CMIP6 data became available for dissemination.

The fully federated ESGF system, now supported by several independently funded groups, has been successfully published and serves over 25 PB of CMIP6 data. Still, it was recognized that, on a tight timeline, ESGF would be expected to serve a CMIP7 phase that was already being planned. A major review of the software system was undertaken with a fundamental reassessment of its requirements, which would guide the development of a new overall system architecture (Kershaw et al., 2020). The redesign was initiated due to challenges around maintaining its existing software and the changing technological landscape. The plans arising from the review were a simplified deployment system, the adoption of broader community standards to facilitate integration with other systems, and the replacement of the data search service.

## 4.5 The IPCC Data Distribution Centre (IPCC DDC)

The Intergovernmental Panel on Climate Change Data Distribution Centre (IPCC DDC) was established in September 1997, at the Thirteenth session of the IPCC in the Maldives. The stated goal was to reduce the barriers to accessing and using CMIP



future scenario datasets, with a particular focus on serving the needs of the IPCC WG2. Since its initiation, this scope has expanded to provide a long-term, persistent archive of the datasets used by IPCC authors indefinitely, ensuring reproducibility.

Since its inception, the IPCC DDC has evolved with numerous contributing institutions and partnerships. In the initial phase, a partnership was arranged between the German Climate Computing Center (DKRZ) and the UK Climate Research Unit (CRU), with the Finish Meteorological Institute (FMI) to contribute guidance and training. The DKRZ took responsibility

for the CMIP model data, and began archiving the AMIP and CMIP model data used in the prior IPCC Assessment Report (SAR, 1995). This work continued in the early 2000s in preparation for the IPCC TAR, with atmospheric near-surface variables collected, aggregated, and disseminated through DKRZ infrastructure (Stockhause and Lautenschlager, 2022).

The scope expanded during the IPCC TAR and following the AR4 period, with additional IPCC WG1 requests to augment the DKRZ archive to include most of the CMIP3 archive, including the new SRES future projection simulations that were made

available at the time (see subsection 2.3). The British Atmospheric Data Centre (BADC) took over the UK contribution formerly provided by CRU. In parallel, the remit of the IPCC DDC expanded to meet the augmented needs of the WG2 and WG3, with a requirement to collate the growing socio-economic data and scenarios that were being developed and underpinned the development of the SRES scenario suite for CMIP3 (Nakićenović et al., 2000). This led to the expansion of IPCC DDC partners, including the US Center for International Earth Science Information Network (CIESIN), based at Columbia University.

Work continued over the AR5 cycle, with the far larger CMIP5 project design (see subsection 2.5) leading to two orders of magnitude growth in the DKRZ reference archive size, comprising 1.7 PB and 910000 datasets at completion, in contrast to 1500 datasets and 1 TB, and 400 datasets and 10 GB for AR4 and SAR/TAR respectively (Stockhause and Lautenschlager, 2022). During this phase, an augmentation of the data quality control procedures was implemented, and Digital Object Identifiers (DOIs) were assigned, making these data citable (Stockhause et al., 2012).

For the AR6 cycle, a considerable augmentation occurred. With the recent development of the Findability, Accessibility, Interoperability, and Reusability guidelines (FAIR; Wilkinson et al., 2016), the additional aim of enhancing the transparency of the AR6 report, especially the traceability of the figures, was added and implemented (Stockhause et al., 2019). Building on the CMIP5 success, DOIs were issued for CMIP6 datasets upon publication to the ESGF (see subsection 4.8). This citation information was embedded alongside the data, made available in the ESGF search results, and used for the CMIP6 data subset

preserved in the DDC long-term archive (Stockhause and Lautenschlager, 2017).

The CMIP6 data augmentation is also linked with improved transparency across observational data products used to produce analyses and figures in the AR6 report.

### 4.6 Infrastructure for the European Network for Earth System Modelling (IS-ENES)

In parallel to the building coordination across institutions and agencies in the US (see subsection 4.4), a similar coordinated

approach was being developed in Europe across CMIP contributing institutions and agencies. The InfraStructure for the European Network for Earth System modelling (IS-ENES) consortium was officially established in March 2009 supported by the European Commission Capacities Programme.



Over three phases of the project, extending from 2009 through 2023, the project coordinated the activities of 30 contributing institutions across 15 countries Many of these projects led or collaborated on the activities summarised in the preceding and following sections (e.g., subsection 4.1 through subsection 4.3, subsection 4.7 through subsection 4.10, and subsection 5.3).

IS-ENES aimed to coordinate European activities across climate modelling, computer science, data management, climate impacts, and climate services and was responsible for much of the coordinated infrastructure work that led to the delivery of the European contribution to the CMIP5 and CMIP6 phases. It led to the development of common models and tools and the efficient use of High-Performance Computing across contributing infrastructure. The project also paved the way for exploiting model data by the Earth system science community, the climate change impact community, and the climate service community, with much of this work continuing through original member institutions today.

## 4.7 Model (and data) documentation

Before model intercomparisons were firmly established community exercises, high-level model documentation was intermittently provided in peer-reviewed publications, with more detailed institutional gray literature reports commonplace. Early in AMIP, it was recognized that improved documentation could help analysts better interpret their results. A first attempt to coordinate model documentation was made available for the AMIP1 simulations on the PCMDI website (Phillips, 1996).

During the planning for CMIP5, the WGCM endorsed an attempt to formally collect model documentation under the auspices of the European "Metafor" project, initiated in 2008 (Guilyardi et al., 2011; Lawrence et al., 2012). Building on work for the US "Curator" project, which began in 2005 (Dunlap et al., 2008), a complex metadata system was designed and built to capture provenance activities in the entire process, from experiment design through model description and execution to data delivery.

The system provided was delivered late and was complex to use, so the system maintainers entered much of the information after CMIP5 rather than the modelling groups during the project. The resulting model descriptions were used for the IPCC AR5 model evaluation chapter (Flato et al., 2013). Many technical lessons were learned during this period, and a new system was designed and built for CMIP6. Unfortunately, it was also delivered late, although the construction of experiment documentation proved helpful (Pascoe et al., 2020).

Without comprehensive model documentation, important initiatives to understand the impact of model structure on simulated climate variability and change have had to rely on a painstaking and incomplete analysis of the available public information (e.g. Boé, 2018) or post-hoc analysis of results (e.g., Masson and Knutti, 2011; Knutti et al., 2013). The need for an improved and simpler system, delivered early so that documentation can be populated during the data publication process to ESGF, is clear.

## 4.8 Data citation

With exponential growth in the climate community using data, the archive of CMIP model output over phases, and the expanding list of CMIP MIPs and their experiments (see Table 1, Table 2), the need to more explicitly assign credit to modelling groups, in addition to uniquely identifying model data used for downstream analyses, tracking errata, and enabling reproducible





science, was needed. Indeed, this was the main request of the modelling groups after CMIP5 (Williams and Lautenschlager, 2016).

For CMIP5, the three primary ESGF nodes at PCMDI (US), DKRZ (Germany), and CEDA (UK), shared the hosting role for the CMIP5 archive - distributed across nodes and not complete at any one centre (subsection 4.4). Data federation led to a
more complex task of uniquely identifying datasets and their downstream downloads and broader use. To solve this problem, DKRZ led an effort to both quality control and assign Digital Object Identifiers (DOIs) to CMIP5 datasets (subsection 4.5; Stockhause et al., 2012), delivered alongside the long-term archival as part of the IPCC Data Distribution Centre (IPCC DDC, subsection 4.5; Stockhause and Lautenschlager, 2022). However, these DOIs were maintained in the IPCC DDC long-term archive managed by the DKRZ, and were not made available alongside the data that was hosted and made available for
download on the ESGF.

Following CMIP5, for CMIP6, a more ambitious plan was established to meet the needs of IPCC DDC archived data and the evolving data real-time (Stockhause and Lautenschlager, 2017). Considering the CMIP project evolution across phases and the federated design that underpinned CMIP6 (see section 3; Eyring et al., 2016a), data citation needed to account for model configurations that were targeted across the 22 MIPs that published data to the project (Table 2). The DKRZ, began a process to
identify and generate CMIP6 DOIs that provided two levels of citation support, leveraging the CMIP6 Controlled Vocabularies (see subsection 4.2; Durack et al., 2024). One targeted the upper level, a single model (source_id) for a unique MIP (activity_id) configuration published in the ESGF CMIP6 archive, and another at the lower experiment (experiment_id) level, allowing users to attribute the dataset used more tightly - identifying a unique dataset matching a model, MIP and experiment.

To date, 2,727 CMIP6 DOIs have been generated, with the first issued in June 2018 to the first CMIP6 dataset published in
May 2018 (see Figure 2). In addition, DKRZ also provided DOI support for the input4MIPs project, minting 210 DOIs for the forcing datasets being used by modelling groups to meet CMIP6 experimental protocols (see subsection 5.1). Currently, more than 6700 recorded citations have been made against these ESGF datasets, approximately the same counts recorded against the CMIP6 overview paper (see section 6; Eyring et al., 2016a).

## 4.9 Data replication

Since the establishment of the federated ESGF system for CMIP5, users have been able to access data through multiple ways. Either through direct downloads using the web-based search interface, or through command-line tools using standard HTTP or, more recently, using the Globus GridFTP protocol, enabling high-throughput transmission of large data volumes - precisely what is needed considering the petabyte scale of recent projects (Table 1).

As the data archives and their construction became more clearly defined through the modern phases (see subsection 4.1
through subsection 4.4), the need for more systematic and reusable tools was required, making data easy to download, and replicate across primary ESGF nodes facilitating the federated CMIP5 and CMIP6 archives.

To serve these needs, the "synchronize data" (or, more familiarly, "Synda") software, developed by the Institut Pierre-Simon Laplace (IPSL) in 2011, was created to facilitate large-scale ESGF data downloads (Denvil et al., 2020). This tool aimed to solve an ESGF evolving software usability problem, which made data discovery and download challenging. Synda,



enabled IPSL to create a mirror of the CMIP5 data serving numerous IPCC authors within Europe who had access to the IPSL systems. When the ESGF software began production in late 2011, with a new SOLR-based index and the esg-search API (https://esgf.github.io/esg-search), synchro-data was updated, becoming Synda by mid-2012.

Challenges accessing CMIP-scale data are rooted in the limitations of traditional download methods, such as manual downloads, point-and-click interfaces, or interactive "wget" scripts. Throughout CMIP5, Synda continuously evolved to address effi-
cient large-scale downloads and meet the demands of high-volume scientific workflows. A key Synda innovation was tracking download histories and only downloading new files matching a predefined query. This allowed the primary ESGF replication nodes, and many institutional infrastructures to replicate CMIP data subsets efficiently. With the addition of GridFTP protocol support, Synda became the official software used by the ESGF Tier 1 nodes, enabling cross-continental CMIP replication and improving archive accessibility.

By 2023, much of Synda's code had become obsolete and needed deep refactoring. The IPSL released the "esgpull" successor in early 2024 (Rodriguez et al., 2024). "esgpull" offers similar functionality with lower latency based on asynchronous technologies and streamlined implementation. It facilitates data managers seeking to replicate data across ESGF nodes and regular users searching and downloading specific datasets. "esgpull" was also integrated into the data synchronisation workflow between ESGF and the subset of climate projections provided to Copernicus Climate Change Service (C3S).

### 4.10 Data downloads

As the infrastructure collating and serving the building multi-model archives became better coordinated, questions began to be raised, how much data was there, how was it distributed across contributing experiments, and by download requests, what were users targeting for download. For CMIP3, this was simple, as a single location and ESG II system hosted the complete archive (see subsection 4.4), statistics were readily available for both the total project data volume (see Table 1) and for total
downloads (see Figure 5), which by the end of the project in 2009 (first data publication occurred in late 2004, Table 1) were well more than an order of magnitude larger than 39 TB total project footprint (470 TB; Ananthakrishnan et al., 2007; Williams et al., 2009).

As the ESGF federated design was realised, in preparation for CMIP5 (see subsection 4.4), the added complexity - no one ESGF node would initially host the data, and users downloading data could download from any node, not just a single one -
brought a new challenge for CMIP infrastructure, how best to quantify project size by modelling group contributions, and the user demands. This new challenge was a focus of discussions at the just commenced annual ESGF international Face-to-Face meetings, and by late 2011, the need for a dashboard documenting the number of published data, downloads and user metrics identified as a high priority improvement for the coming months (Williams, 2011).

With the support of IS-ENES (see subsection 4.6) in collaboration with DoE, ESGF contributors from the Italian CMCC
Foundation (Euro-Mediterranean Center on Climate Change) developed a metrics and dashboard capability. By 2015, the first version of the ESGF dashboard was live, providing project statistics for CMIP5, in addition to the smaller supporting projects obs4MIPs (see subsection 5.2) and CORDEX-CMIP5 (Williams and Lautenschlager, 2016). This service captures a comprehensive set of data usage and archive metrics for a single ESGF node, and at the cumulative federation level. The data





usage information provides the number of successful downloads and distinct downloaded files, with a granularity that enables
an interrogation of these metrics by project, variable, institution, model, and experiment. In addition, data archive information
is available per project, quantifying the total number of published datasets, data volume, and granular information related to
the CMIP project's contributing models and institutions.

Today, the CMCC ESGF Dashboard continues to provide metrics across the ESGF projects: CMIP6, CMIP5, CORDEX-
CMIP5, input4MIPs (see subsection 5.1) and obs4MIPs (Fiore et al., 2019). The service collates project data and downloads
statistics from a subset of the total contributing nodes across projects (see Table 1). For the CMIP5 project, 7 ESGF nodes
are currently reporting data and download statistics, and for CMIP6, 9 nodes are reporting, with temporal coverage from 2018
through to the present. This is a subset of the total 30 nodes currently (CMIP6) or were (CMIP5) publishing data to the archive.

The CMCC-provided dashboard statistics (2018-), in addition to archived ESG II statistics from the single LLNL/PCMDI
CMIP3 project portal (2004-2009), have been used in the preparation of Figure 5.

## 4.11 MIP errata

The growing coherence across successive CMIP archives allowed for more complete data tracking and the development of an
errata list documenting user-reported problems. For CMIP3 and CMIP5, user reports via email were captured by PCMDI, and
the tabulated information was hosted on an external-facing website and reported to the relevant modelling centre. The slow
process yielded a collated error list representing issues across the archive, which led to several data revisions (see Table C1).
For CMIP3, user-reported errors led to the retraction of all simulations from a single modelling centre, as a significant bug was
found in the model formulation, rendering its output obsolete.

Simultaneously, many modelling centres maintained their inventories of problems affecting their outputs, ranging from
minor issues to critical errors that could compromise data usability. However, these inventories were not always public, and
when they were, they were often not easily accessible and regrettably not co-located with the user-reported issues.

Building on its experience in organising and tracking issues with its datasets, the Institut Pierre-Simon Laplace (IPSL)
decided after CMIP5 to develop an errata database, providing a standardised interface to document both user-reported and
modelling centre-reported issues. In late 2013, a proof-of-concept was developed and proposed to the ESGF community. To
enhance documentation, the initial version of an operational errata system was implemented within the ES-DOC (Earth System
Documentation; see subsection 4.7) ecosystem in 2015 (Pascoe et al., 2020).

This service was significantly enhanced for CMIP6, leveraging the Controlled Vocabularies (see subsection 4.2; Durack
et al., 2024). In June 2018, a new version hosted at IPSL was launched, designed to be project-agnostic and support CMIP6,
CMIP5, CORDEX-CMIP5, and input4MIPs data from the outset. The service introduced a centralized hub for documenting
issues, with a standardised issue creation form to ensure consistently harmonised reporting across modelling centres. This
standardisation, combined with automated data checks and integration with third-party tools such as quality assurance systems
and persistent identifier frameworks, made the service highly interoperable and adaptable to various workflows. In addition,
the system also allowed for the issue severity to be identified, partitioning minor metadata issues from more severe problems
that could impact a downstream analysis. For CMIP6, more than 460 issues have been identified to date, with more than half





of these resolved by the responsible modelling group by providing additional metadata or rectifying and recreating the errored dataset.

While the service gained traction, its adoption was somewhat limited compared to the scale of CMIP6 datasets, particularly concerning the number of retracted versions (less than a quarter of the modelling groups taking part in CMIP6 used the errata). This outcome reflects the voluntary nature of errata documentation and the time constraints faced by those responsible for its implementation (typically data managers within modelling centres).

Further developments led to the release of version 2.0 in July 2023. This version introduced the ability for all users to
contribute errata entries for issues identified in the CMIP and CORDEX data catalogues, with submissions moderated by the corresponding data providers/modelling group representatives. By incorporating crowdsourcing, this approach aims to enhance coverage across the data pool while ensuring documentation quality through moderation.

## 5    CMIP6 supporting projects

In addition to the core infrastructure that has delivered past CMIP phases, several uniquely identifiable supporting projects
have developed over the past decades. Below is a brief overview of a number of the more clearly defined activities that have served CMIP6 or preceding MIP phases.

### 5.1    input4MIPs: climate forcing for MIP experiments

As the scientific scope and experiment count broadened through MIP phases (Figure 1, Table 1), the need for climate "forcing" datasets to guide simulation evolution continued to increase. For the initial AMIP1 experiment (1979-1988; AGCM), green-
house gas concentration and solar irradiance were preset as fixed fields ($CO_2$ 345 ppm, and 1365 $Wm^{-2}$ respectively), with time-varying sea surface temperature (SST) and sea ice boundary conditions imposed. As model and experiment complexity increased, more and more complex suites of forcing agents needed to be collated for modelling centre use.

AMIP2 incorporated additional greenhouse gases, ozone, aerosols, and more complex interactive land components (e.g., Gleckler, 1996; Liang et al., 1997). In CMIP1 and CMIP2/CMIP2+, an interactive ocean and sea-ice removed the need for SST
and sea-ice boundary forcing, extending the experiment end date to 2001 (see Table 1).

With the definition of the CMIP3 20C3M/historical experiment ($\sim$1850-2000; Meehl et al., 2007a), considerable augmentation occurred, with the first historical period datasets hosted by PCMDI alongside existing early AMIP/CMIP data. These data mainly addressed the SRES future scenarios (2001-2100/2200/2300; Nakićenović et al., 2000), but did include a small number of historical period forcings including those that had already been collated for the AMIP2 experiment period (1979-2001;
Table 1). Additional forcing needs were met independently by each contributing modelling group and model configuration.

For CMIP5, increased model complexity (many models included atmospheric chemistry by this stage) necessitated a far more comprehensive approach to meet the historical experiment forcing needs (1850-2005; Taylor et al., 2012b), and to share the workload, considerable coordination across modelling groups occurred. For anthropogenic emissions of short-lived climate forcers (SLCFs 1850-2000; e.g., CO, NOx, etc), dataset creation was led by NCAR, with contributions from a broad community



including observationalists and climate modellers (Lamarque et al., 2010). This effort was also extended to cover nitrogen and sulfur deposition over the historical and future period (e.g., Lamarque et al., 2013). A similar community effort was taken with tropospheric and stratospheric ozone, led by the German Deutsches Zentrum für Luft- und Raumfahrt, Institut für Physik der Atmosphäre (DLR; Cionni et al., 2011). Well-mixed greenhouse gas concentrations (WMGHGs; 1765-2300) were also community-developed, however, this dataset development was dovetailed with the generation of the RCP future scenario data

extending out to 2300 and consequently also entrained and even broader Integrated Assessment Modelling (IAM) community tasked with scenario development (Meinshausen et al., 2011). The concentrations and the anthropogenic emission datasets were developed to span both the historical and future projection periods, which for the first time necessitated harmonizing these quantities across the historical-RCP 2005-2006 transition. Additional dataset development contributed to the project fell back to an existing community of researchers involved in forcing production for early phases.

However, during CMIP5, there was limited standardisation of forcing datasets, particularly those used in the historical and piControl experiments, which differed markedly across modelling groups due to the lack of a centralised and coordinated approach. As an example of forcing differences across the historical simulations, the NOAA-GFDL GFDL-CM3 (Donner et al., 2011) used solar variability (Fröhlich and Lean, 2004), volcanic stratospheric aerosol emissions (Stenchikov et al., 2006) and land-use changes (Hurtt et al., 2006). Whereas NASA-GISS GISS-E2 (Miller et al., 2014) used solar variability (Wang

et al., 2005), with spectral variations (Lean et al., 2003, updated 2009), volcanic stratospheric aerosol emissions (Sato et al., 1993, updated), and land-use changes (1850-1900; Pongratz et al., 2008) and HYDE3.0 (e.g., Klein Goldewijk et al., 2011) thereafter.

For CMIP6, it was recognised that forcing dataset coordination was needed. The input datasets for model intercomparison projects (input4MIPs) ESGF project was established in 2016, and the collation of forcing data required for the CMIP6 DECK

experiments commenced. In addition, the collation of additional data supporting 14 other Community MIPs was also published and made available to participating groups (Durack et al., 2018). The input4MIPs project collated next-generation forcing datasets extending from 1850 to 2014 for SLCFs (Hoesly et al., 2018; Feng et al., 2020), biomass burning (e.g., fire; van Marle et al., 2017), land use changes (Hurtt et al., 2020), WMGHGs (Meinshausen et al., 2017), volcanic stratospheric aerosol optical depth (e.g., Thomason et al., 2018), ozone (Checa-Garcia et al., 2018), nitrogen deposition (Hegglin et al., 2016), solar

irradiance (Matthes et al., 2017), SSTs and sea ice boundary conditions (Durack and Taylor, 2018) and anthropogenic aerosol optical properties (e.g., Fiedler et al., 2019).

## 5.2   obs4MIPs: observations for model evaluation

The fundamental role of large-scale, gridded observational products for climate model evaluation was recognized before the advent of organized MIPs. It became a crucial part of the AMIP1 (Gates, 1992a) and AMIP2 (Gleckler, 1996) phases, with a

heavy independence on the ECMWF and NCEP/NCAR reanalysis products at that time.

Modelling groups and analysts continue collecting observational data for their specific needs, typically ad-hoc, with limited coordination.





The observations for Model Intercomparison Projects (obs4MIPs) activity was established to facilitate model evaluation via the technical alignment of gridded observational products with CMIP standards (see subsection 4.1 and subsection 4.2). It is ac-
complished by metadata/data integration across projects, facilitating side-by-side data delivery through ESGF (subsection 4.4). The project originated from a PCMDI and NASA collaboration (Gleckler et al., 2011) and was later internationalized by the WCRP Data Advisory Council (WDAC; Teixeira et al., 2014). An international workshop helped identify critical pathways to improve the effort in preparation for CMIP6 (Ferraro et al., 2015). Many of these recommendations remain relevant and are being addressed today, for example, with the inclusion of higher frequency, process-relevant gridded data and in-situ station
measurements, and a detailed obs4MIPs progress update during CMIP6 was published (Waliser et al., 2020).

During the late stages of CMIP6, obs4MIPs was revitalized (https://pcmdi.github.io/obs4MIPs), with efforts striving to ensure that obs4MIPs can accelerate model evaluation, research, and development in future phases of CMIP and related modelling activities.

### 5.3 Coordinated climate model evaluation

AMIP made model output available to analysts with diverse expertise, allowing for more varied and in-depth model evaluation. It established a benchmark against which new and possibly improved models could be evaluated, emphasizing the simulated mean climate and variability assessment on large to global scales. A primary goal was to systematically document errors across a suite of similar models, which had never been done before, and to advance and improve the simulation of physical processes and phenomena (Gates, 1992b). Recognizing the value of a community-based approach to routine model evaluation,
the WGNE created the Standard Diagnostics of the Mean Climate, to be used in AMIP/CMIP, and in parallel, the Standard Diagnostics of Variability in use by numerical weather prediction (e.g., WMO World Climate Research Programme, 2003, p. 10). The community definition of model evaluation diagnostics, naturally led to an augmentation of the standard output requested to meet these analysis needs (see subsection 4.3).

By 1994, this philosophy was the explicit science driver behind CMIP1, "intercomparison makes for a better climate model"
(Meehl et al., 1997). Since the early days, much progress has been made, with "systematic community evaluation" achieved across every MIP phase since FANGIO. The most prominent examples are the dedicated climate model evaluation chapters that have featured in every IPCC Assessment Report since the FAR in 1990, targeting evaluation of the model generation of the corresponding MIP phase (e.g., Gates et al., 1990, 1996; McAvaney et al., 2001; Randall et al., 2007; Flato et al., 2013; Eyring et al., 2021a). As model and evaluation complexity increased, so has the chapter content, expanding from 39 to 130
pages FAR to AR6 (Gates et al., 1990; Eyring et al., 2021a), or including the AR6 model annex (Gutierrez et al., 2021) and supplementary (Eyring et al., 2021b), 359 pages (noting AR6 Chapter 3 also covered climate change detection and attribution, which in previous reports had a dedicated chapter).

The development of internationally coordinated MIP activities naturally nurtured collaborative community evaluation by the active participants, a feature of many activities running parallel to CMIP (e.g., C4MIP, CFMIP, PMIP, etc; Carbon-Land Model
Intercomparison Project (C-LAMP): Hoffman et al., 2007; Ocean Carbon-Cycle Model Intercomparison Project (OCMIP): Orr, 1999; Dutay et al., 2002, 2004).





Outside the IPCC model evaluation chapters, the concept of coordinated evaluation by MIP non-participants was uncommon. With the open access and enhanced standardisation of CMIP3 (see subsection 2.3, section 4), in addition to the availability of model simulations that were directly comparable to the coincident observational record, this led to more routine and

comprehensive evaluation being undertaken. The project advancement nurtured a more diverse and extensive data user group engagement, leading to several seminal contributions that previously would not have been possible (e.g., Eyring et al., 2006; Gleckler et al., 2008; Waugh and Eyring, 2008) and which were heavily cited in the subsequent IPCC AR5 model evaluation chapter (Flato et al., 2013).

Through CMIP5, this progress continued, with the aspiration of climate model systematic evaluation leading to the develop-
ment of several dedicated analysis packages targeting the CMIP DECK (Eyring et al., 2016b). The ESMValTool (Earth System Model Evaluation Tool; Eyring et al., 2016c; Righi et al., 2020; Eyring et al., 2020) is a coordinated effort that supports the analysis and evaluation of Earth system models by collating a wide array of observational datasets and defining diagnostics that allows for the direct comparison of model simulation output with observations. This effort grew out of the precursor CCMVal-Diags (Chemistry-Climate Model Validation Diagnostic tool; Gettelman et al., 2012), which had channelled earlier
work (Eyring et al., 2006; Waugh and Eyring, 2008) into reproducible analysis. The PCMDI Metrics Package (PMP) is another key tool, which built on the pioneering work from CMIP3 (Gleckler et al., 2008), providing a standardized climate model evaluation framework, comparing their outputs with observations curated by the obs4MIPs project (subsection 5.2) across a range of space- and time-scales and variables focused on physical climate mean states and their modes of variability (Gleckler et al., 2016; Lee et al., 2024). The ILAMB (International Land Model Benchmarking) package grew out of the earlier C-LAMP
activity (Hoffman et al., 2007) and is focused on assessing the land surface sub-models, offering comprehensive diagnostics and benchmarking to evaluate land-atmosphere interactions (Collier et al., 2018). Additional packages that also facilitate and leverage community engagement include the climate process-oriented NOAA Model Diagnostic Task Force (NOAA-MDTF; Neelin et al., 2023), and NCARs Climate Variability Diagnostics Package (CVDP; Phillips et al., 2014), amongst numerous others.

These and other community-developed tools (e.g., Hoffman et al., 2025) will play increasingly pivotal roles in ensuring climate models are useful tools for climate change prediction and continue to improve through rigorous and systematic community evaluation. It is likely, as the use of CMIP output continues to expand into climate change impacts and adaptation planning, that these tools will be more and more heavily utilized to evaluate simulation data fitness-for-purpose and potentially lead to more advanced methods for model selection and weighting (Eyring et al., 2019).

**6  CMIP impact**

By establishing benchmark experiments along with innovative targeted experimentation, and by facilitating access to the results, CMIP and the other MIPs relying on its infrastructure have fundamentally changed the expectations and research practices in climate modelling. It has become common practice to assess areas where models should not be trusted by considering where



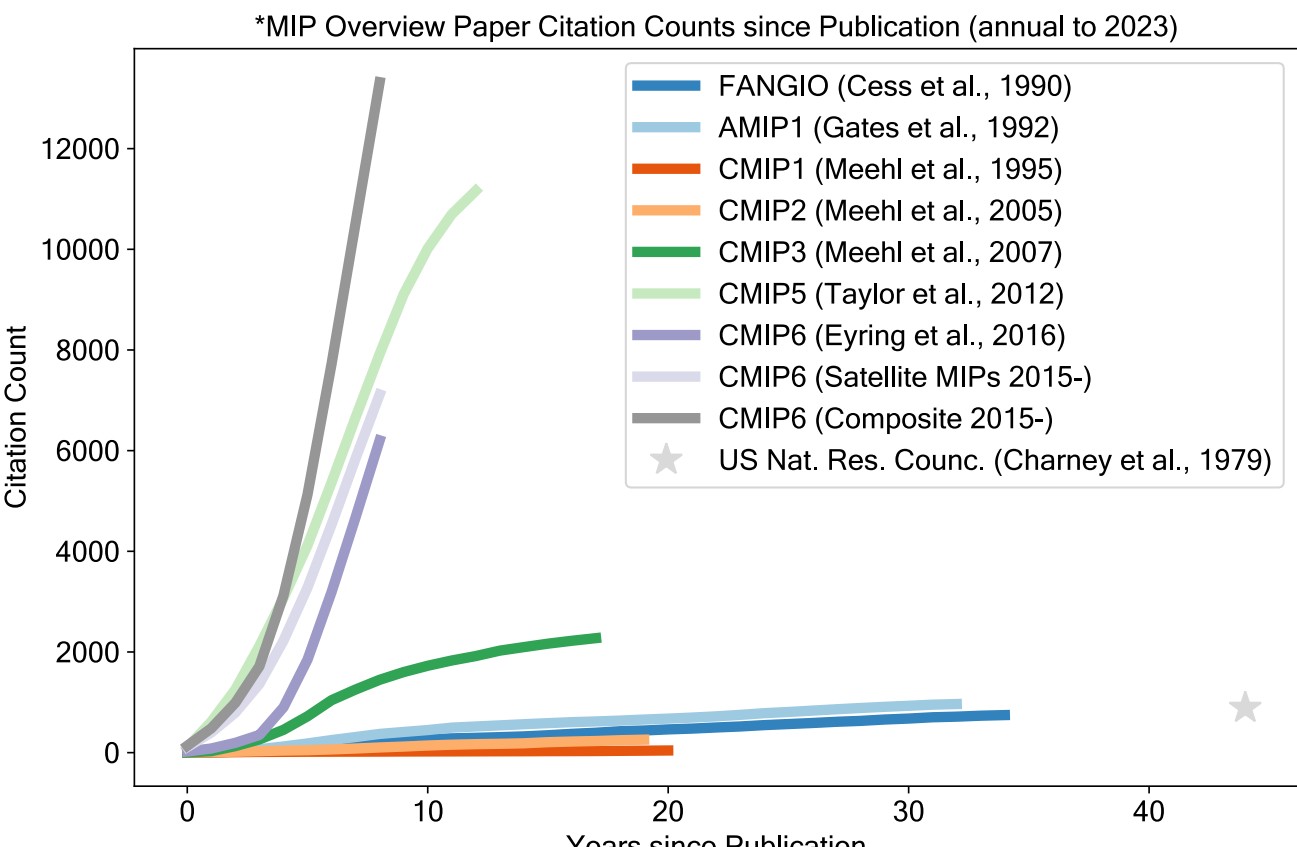

**Figure 3.** Web of Science (WoS) citations for key overview papers of each MIP phase, from FANGIO (Cess et al., 1990) through CMIP6 (Eyring et al., 2016a), plotted relative to their year of publication (WoS records queried 9 December 2024). For CMIP6, citations are plotted for the project overview paper (Eyring et al., 2016a), for the cumulative total across each of the satellite Community MIPs that comprise the phase (see Table 2), and for the composite sum of these two sources. Citation counts are an imperfect way to capture MIP impact but reflect a large and strongly growing user community using MIP data across numerous phases.



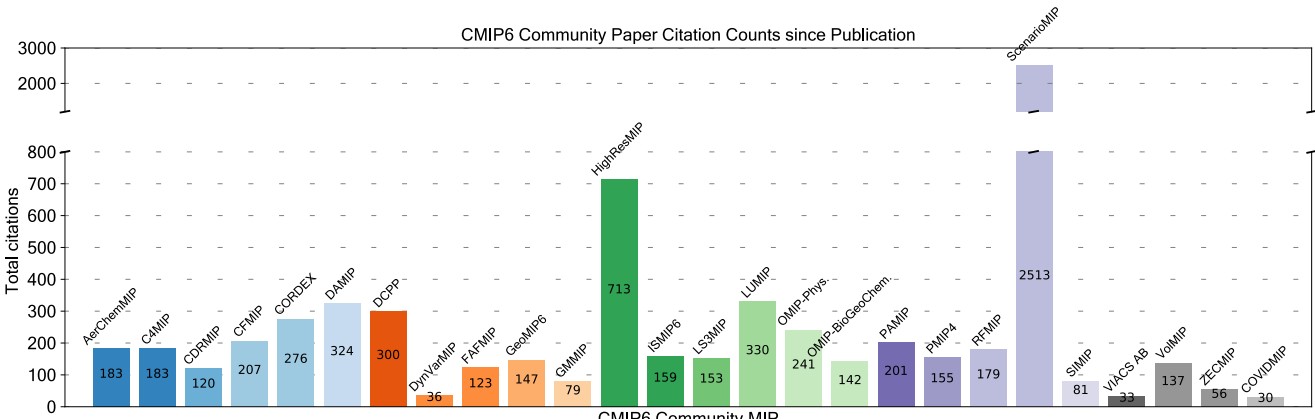

**Figure 4.** Web of Science (WoS) citations for the 23 endorsed MIP overview papers contributing to CMIP6 (see Table 2) from their publication to today (WoS records queried 9 December 2024). Black text numbers on each coloured bar denote citation counts. Citation counts are an imperfect way to capture MIP impact, however, reflect the large interest in the ScenarioMIP future climate change projections (O'Neill et al., 2016) and the HighResMIP high-resolution simulations (Haarsma et al., 2016).

their results diverge. Although agreement among models does not imply their results are correct, the multi-model perspective generally enables a richer interpretation than the results from a single model.

During its four decades of operation, the infrastructure supporting MIPs has been essential in building a large and expanding collaborative community (e.g., Figure 1, Table A1). However, this breadth and expansion make it challenging to gauge CMIP's impact quantitatively, identify the breadth and diversity of downstream users, and apportion recognition to its contributors and participants. The move to an open data paradigm in CMIP3 (and to open data licenses, Table 1) has complicated the interpretation of data downloads as a measure of interest and impact of the project. This problem is compounded by the creation of secondary data repositories where users can download/access data, outside of ESGF's licensing, access control, and monitoring (e.g., Balaji et al., 2018). In addition, compute services co-located with the primary ESGF replication nodes (e.g., PCMDI, DKRZ, and CEDA), other regional nodes (e.g., NCAR, NCI) and the replication of datasets into the commercial cloud (e.g., PANGEO; Abernathey, 2020), which serve an even larger and more diverse online community, further complicate the assessment of impact based on available data downloads and use statistics (as no downloads are logged for repeat uses of in-place data).

CMIP data are now routinely used outside the physical climate science community and, more broadly, outside the academic community. This is a dramatic shift from the early days of FANGIO/AMIP, which began the climate model intercomparison activities and involved the modelling group participants alone. Consequently, academic literature citations are an imperfect metric for quantifying impact. However, since a representative "overview" paper exists for each phase, it is useful to use the citations of these papers to evaluate the relative impact of each MIP phase (see Figure 3). Similarly to the marked project growth as captured in Figure 1, the project impact, measured through academic citation, has grown dramatically (Figure 3).





To place in context the citations plotted in Figure 3, the seminal work of the Charney report (Charney et al., 1979) has been included. This report relied on idealised climate change predictions from three climate models (two US, NOAA-GFDL, and NASA-GISS, and one UK, MetOffice). It is the beginning of many of the targeted US climate change research efforts (see section 2). The Charney report was published by the US National Academy of Sciences, and consequently, time-evolving Web of Science citation counts are not available; instead, only a single cumulative citation count from Google Scholar (GS) of 889 is available (see Figure 3).

As evident in the figure, starting with CMIP3 (Meehl et al., 2007b) and its open and standardised data approach, CMIP saw far greater interest than the preceding AMIP1/2 and CMIP1/2/2+ phases, with a total of more than 2200 Web of Science (WoS) citations and more than 3400 GS citations to date. Building on CMIP3 momentum, CMIP5 saw an even more dramatic step change in interest. To date, CMIP5 (Taylor et al., 2012b) has received more than 11000 WoS and more than 15000 GS citations.

For the most recent phase, CMIP6, the momentum continues to build with more than 6000 WoS and 8000 GS citations (Eyring et al., 2016a). If we augment these counts to include the 26 articles that define each CMIP6 Community MIP (see Table 2), WoS already exceeds 13000 (CMIP6 Composite 2015-, Figure 3).

The CMIP5 (Taylor et al., 2012b) and CMIP6 (Eyring et al., 2016a) papers are both identified as WoS "highly cited papers," ranking them in the top 1% of Geosciences publications.

Citations of CMIP6-endorsed MIP experiment design papers can be used to gauge interest in individual MIP activities and their experiments (Figure 4). These results show very strong interest in two primary activities: the future-focused ScenarioMIP (O'Neill et al., 2016) and the high-resolution-focused HighResMIP (Haarsma et al., 2016), which received more than 2400 and 700 WoS citations, respectively. Before CMIP6, MIP experiment design papers often appeared in the gray literature, so a systematic analysis of citations over time is impossible. However, it can be claimed that the seminal science enabled by MIP results grew as momentum was built behind the MIP paradigm.

An alternative, although similarly imperfect, impact metric is based on data download statistics. Like citation counts, these provide relevant measures of user interest across CMIP phases, across contributing or endorsed participating MIPs, and across their respective experiments. An advantage of download assessment is that it should capture interest outside the academic community, which may be less inclined to record its use of data through peer-reviewed publications. Before CMIP3, however, no data download records are available, which limits a comprehensive assessment. An additional complexity is that direct comparison across phases is impossible due to the marked increase in activities and experiments over time (see Table 1, Table A1).

Like the literature citations, a clear picture emerges when the download data is assessed comparably across phases (Figure 5). These show an evident repeating dominance of three key experiments, the CMIP/DECK 20C3M/historical (dark orange) and piControl (light orange), and all flavours of the ScenarioMIP experiments (which have been cumulatively pooled, red; CMIP3: 3 SRES; CMIP5: 4 RCPs; and CMIP6: 8 SSPs) dominating downloads. For each of the successive phases, 6 of 12 CMIP3 experiments account for 94.3% of downloads (top), 10 of 37 CMIP5 experiments account for 90.3% (middle), and 16 of 322 CMIP6 experiments account for 90.7% of downloads (bottom). Considering only the top three experiments across phases



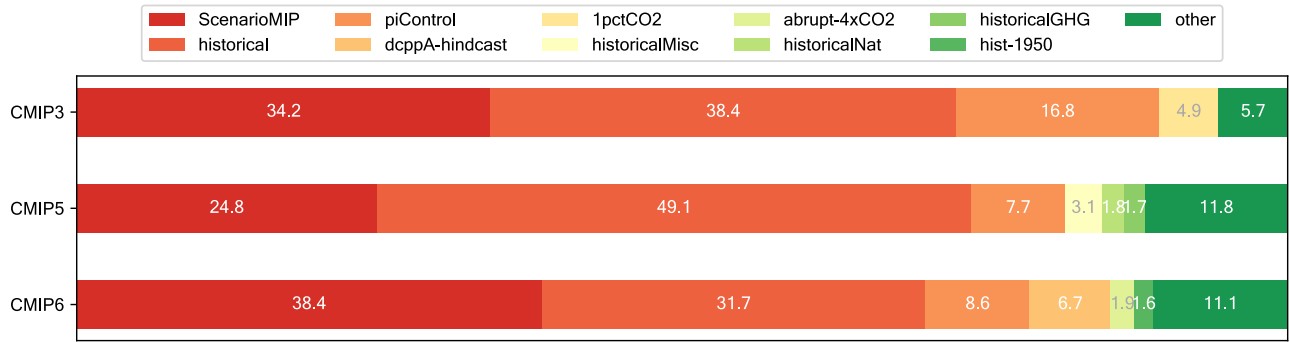

**Figure 5.** Downloads across the three most recent phases of CMIP, for which records are available, presented as a percentage total. Each horizontal stacked plot represents total download counts as percentages from 2018 to 28 November 2024 for CMIP5 and CMIP6, and 2004-2008 for CMIP3. For display, all ScenarioMIP experiments (CMIP3: SRES; CMIP5: RCP; CMIP6: SSP) are presented as cumulative totals across all contributing experiments, with other results displaying single experiment downloads. Top, the CMIP3 (Meehl et al., 2007a) downloads across the top 6 experiments (3x ScenarioMIP) of 12 experiments total (see Table A1), representing 94.3%. Middle, the CMIP5 downloads across 10 experiments (4x ScenarioMIP), representing 90.0%, of 37 experiments that defined the phase (representing 8 MIPs identified in Table A1; Taylor et al., 2012b). Bottom, the CMIP6 downloads across 17 experiments (11x ScenarioMIP), representing 90.4% representing 20 MIPs that published data to ESGF (see Table 2). For CMIP6, almost 45% of downloads were accounted for by the 12 core CMIP/DECK simulations (Eyring et al., 2016a), with 38% accounted for in the 11 ScenarioMIP future projection experiments (O'Neill et al., 2016). This pattern of data download priority is mirrored across the prior phases, CMIP5 (CMIP 61%, ScenarioMIP 25%) and a more dominant 20C3M/picntrl demand in CMIP3 (CMIP 64%, 35% ScenarioMIP).

(with the downloads of all ScenarioMIP experiments pooled), these account for 89%, 80%, and 76% of the total downloads for CMIP3, CMIP5, and CMIP6, respectively. A similar assessment can be made when considering the same download records grouped across activities, with CMIP/DECK and ScenarioMIP again dominating totals (Figure A1).

In an attempt to document the broader climate science impact, chronologically aligned with the MIP phases, Figure 6 displays an approximate time-history of MIP activities and other milestones noted in earlier sections, along with the IPCC assessment attribution statements and international agreements that were made. For each IPCC assessment, the calibrated language likelihood scale (e.g., Mastrandrea et al., 2010) assessment of human influence on climate is noted, beginning with "humankind is *capable* of raising the global-average annual-mean surface-air temperature" in FAR (IPCC, 1990) through to the definitive "It is *unequivocal* that human influence has warmed the atmosphere, ocean and land" in AR6 (IPCC, 2021).

There is sometimes confusion regarding the relationship between CMIP and the IPCC assessments, with some thinking that CMIP and IPCC are interchangeable or that CMIP is somehow "run" by the IPCC. This is, of course, not the case. CMIP (and AMIP before it) was formulated as a scientific research activity whereby the modelling groups performing present-day and future climate simulations could intercompare their results to advance understanding of the climate system and publish their findings in peer-reviewed scientific papers. The IPCC assessments depended directly on the papers emerging from the



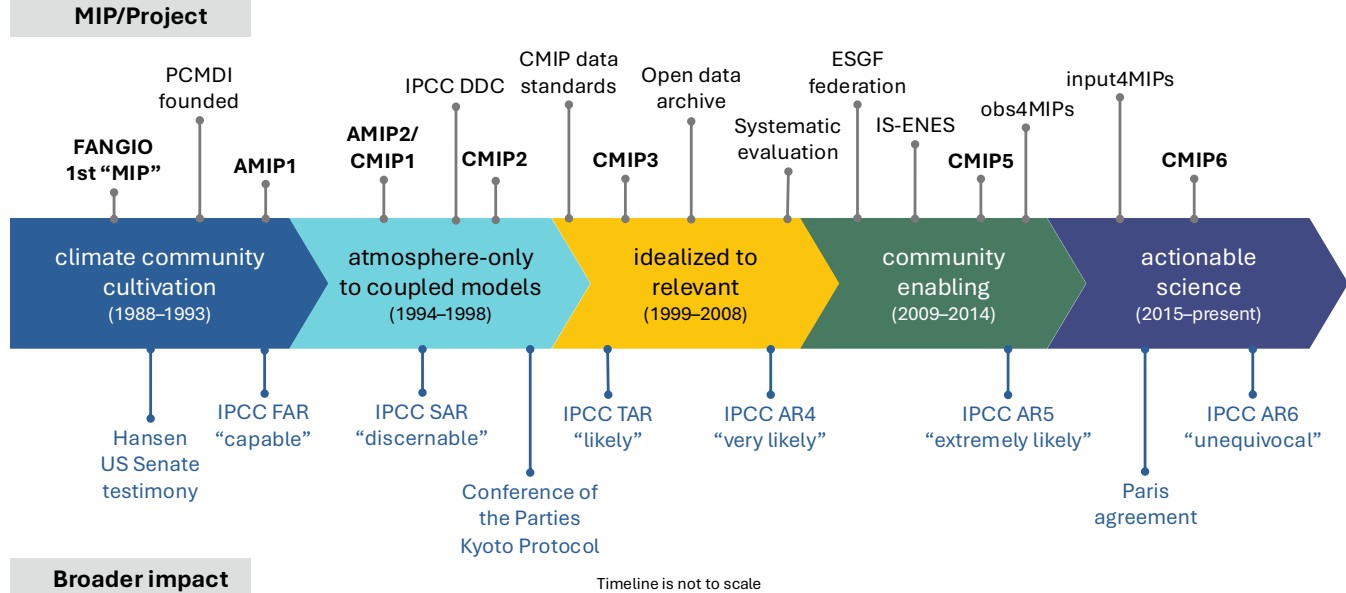

**Figure 6.** A time history of MIPs and their broader impact, with particular relevance to the IPCC Assessment Report phases and statements of human influence on the climate (in parentheses, FAR through AR6; see subsection 2.1 through section 3).

CMIP phases to formulate periodic updates of the current understanding of climate variability and change. However, as the IPCC assessments grew in importance, the modelling groups participating in CMIP were aware of the IPCC process, which motivated, in part, their participation in CMIP and determined the timelines established for each of the latter CMIP phases

(CMIP5, CMIP6). Thus, a symbiotic relationship developed between CMIP and the IPCC assessments (Meehl, 2023). However, without CMIP, the IPCC assessments could not have been possible. Without the coordinated community climate science efforts embodied by the AMIP and CMIP phases, progress in Earth's climate understanding would not have advanced to our present state of knowledge.

# 7   CMIP phase 6 completion

The CMIP6 project is now mostly complete, with nearly all CMIP modelling groups prioritizing ongoing model development over running CMIP6 simulations. Consequently, the growth of the ESGF CMIP6 archive has markedly slowed (see Figure 2).

CMIP has realized the potential of open-access data, enabling science discovery and reproducibility. It has become a de facto standard and an umbrella project for the distributed special-interest Community MIPs to organize their science and cross-institutional collaborations. More generally, the CMIP-supporting infrastructure is increasingly being relied upon to

1050 facilitate and enable coordinated climate science. Any CMIP-contributing modelling group can design an experimental protocol to address some scientific question and, through existing relationships and connections, engage like-minded researchers in collaborating modelling centres to tackle the problem using a multi-model framework (e.g. Jones et al., 2024).





After nearly all CMIP6 simulation results had been published and the IPCC AR6 had been published, the WGCM and the CMIP and WIP Panels undertook a CMIP6 survey to assess project success and gather community feedback (O'Rourke, 2024). For the CMIP-supporting infrastructure, this survey identified some clear priorities, acknowledging CMIP6 progress, but calling for further improvements. In particular, it called for the "CMIP framework", including all supporting infrastructure (see section 4, and section 5) to be maintained and made available in an ongoing capacity, so that this building infrastructure could continue to serve the growing CMIP contributor and downstream user communities.

An interim phase, CMIP6Plus, was initiated to address this need and expand the now well-established data standardisation requirements, data-sharing culture, and collaborative goodwill proven essential to CMIP's success (Mizielinski et al., 2024). The project, led by the WIP, has continued to develop the underlying infrastructure responsible for delivering CMIP6, adapting and modularising it to enable ongoing use with limited additional technical investments by contributing modelling groups or MIP leads. The goal has been to establish consistent data requirements and a sound supporting infrastructure serving to minimise data preparation and publication efforts and enhance scientific productivity and impact. This allows several CMIP6 follow-on activities to continue leveraging the infrastructure in the service of climate science research. The project provides ongoing, but limited support for coordinated model experimentation.

Planning for the 7th phase of CMIP (CMIP7) has now begun with an emphasis on broad community consultation (O'Rourke, 2024). As discussions continue on the core science foci of CMIP7 (e.g., Dunne et al., 2023, 2024), CMIP infrastructure providers are undertaking work (e.g., Kershaw et al., 2020) to modernise and ready services for a relatively small number of CMIP7 AR7 "Fast Track" experiments, which are expected to begin in mid-2025 (Dunne et al., 2024). At that time, new additions to the CMIP6 data archive will cease, and the next-generation simulations based on the latest model versions, updated forcing datasets (subsection 5.1), and refined experimental protocols may begin to be published.

## 8  Summary

CMIP6 is the latest in a long history of internationally coordinated and scientifically collaborative climate model-based research projects. CMIP has developed standard approaches to evaluate and intercompare climate models and a standardised vocabulary and infrastructure for defining and delivering data to a broad and expanding community. It has ensured that projections of future climate conditions are based on a robust and consistent framework.

It has now been 35 years since a group of experts with an interest in modelling standards and climate model intercomparison met informally in Boulder, Colorado, a meeting that led to the first MIP, AMIP1 (Gates, 1991). Since then, MIPs have captivated and engaged a broad and growing number of researchers who have had a tangible impact in improving our knowledge of Earth's climate system.

CMIP has generated profound scientific insights that define how we understand and address climate change and our ability to quantify and attribute the drivers and responses to the observed climate changes we are experiencing today. It has improved climate prediction, provided quantified insights that guide policy and decision making, informed risk and adaptation strategies





and climate change mitigation planning, and improved public awareness and climate education. Its contributions have touched almost every aspect of society and raised awareness of the urgent need for global action on climate change.

Although the project has been a demonstrable success with its focus on the dual goals of facilitating cutting-edge climate research and delivering climate data that enable a broad array of downstream activities, CMIP's growth and broadening community expectations have strained existing resources traditionally devoted to the priorities of modelling centre research staff.

There is some concern that the value of CMIP in meeting the climate information needs of those outside the research community is draining research funds that might be used more productively to develop better models and carry out innovative research. This pressure is not new. After CMIP3, there were calls for a reformulation of efforts to centralize resources across contributors to produce very high-resolution, regionally relevant climate predictions (Shukla et al., 2009, 2010). Somewhat similar calls have been repeated near the end of this most recent CMIP6 phase (Jakob et al., 2023; Stevens, 2024).

Although modelling groups continue to shoulder most of the responsibility for CMIP, the WCRP created in March 2022 a CMIP International Project Office (CMIP-IPO) funded and hosted by the European Space Agency (ESA, UK) to assist the CMIP Panel and the WIP in coordinating the project. The CMIP-IPO is tasked with supporting the design and development of the upcoming CMIP7 project (see https://wcrp-cmip.org/cmip7) and facilitating broad and growing engagement of individuals that might participate in it and benefit from its results. CMIP7 is expected to meet the needs of the IPCC AR7 cycle through

a continued federation of activities that will benefit from the lessons learned in CMIP6 (Eyring et al., 2016a). Efforts will be made to reduce modelling group burdens by better consulting the broad community, clarifying needs, and apportioning the limited resources to meet them.

The CMIP project has had a sustained global impact. Its success has depended on the efforts of tenacious individuals or small teams that encouraged and facilitated initial community engagement. These initial steps, coupled with the coordinated

efforts within and across modelling groups, the infrastructure providers, and other contributors, led to the broad community collaboration and coordination which resulted in CMIP's ongoing impact. At its core, the project is a fundamental anchor point of international climate research, facilitating the generation of climate simulations of value to research and climate change planning. In return, modelling groups benefit from coordinated, collaborative activities and community evaluation, which feed back on the climate model development process, suggesting that indeed "intercomparison makes for a better climate model."

*Code and data availability.* Data underpinning figures in the paper, in addition to tabulated additional information can be viewed in a paper-dedicated GitHub repository at https://github.com/durack1/CMIPSummary, or online using NBViewer at https://nbviewer.org/github/durack1/CMIPSummary/blob/main/figuresAndTables.ipynb.

## Appendix A: Defined experiments across MIP phases

There is considerable continuity with experimental protocols across phases. The "amip" AGCM experiment is where MIP
science began, covering the 1979-1988 period for AMIP1, and 1979-2001 for AMIP2. The concept of a fixed climatological



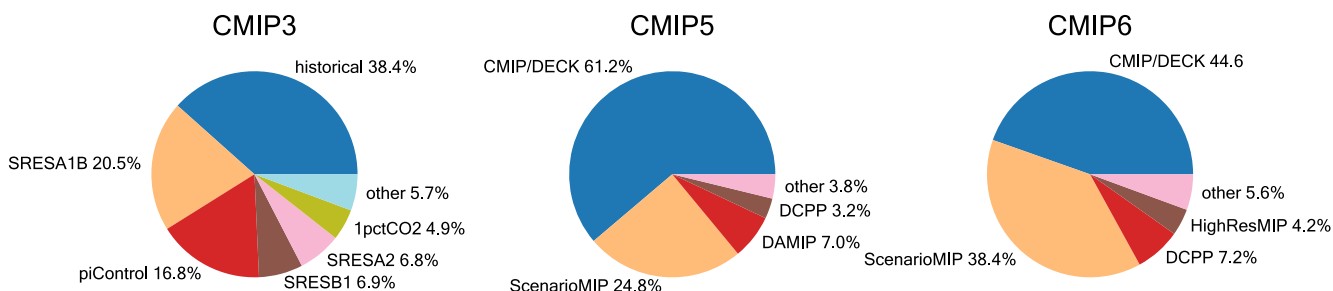

**Figure A1.** Recorded downloads across the three phases of CMIP, for which download records are available, following Figure 5. Left the top 6 CMIP3 experiment downloads across the 12 experiments that defined phase (Meehl et al., 2007a). Middle the CMIP5 downloads for the top 4 MIPs of the 8 that defined the phase (Table A1; Taylor et al., 2012b). And right the CMIP6 downloads for the top 4 MIPs of the 22 that defined the phase ([see Table 2; Eyring et al., 2016a]. For CMIP6, almost 45% of downloads were accounted for by the core CMIP/DECK simulations, and 38% accounted for in the ScenarioMIP future projection experiments (O'Neill et al., 2016). This pattern of data download priority is mirrored across the prior phases, with CMIP5 (CMIP 61%, ScenarioMIP 25%) and a more dominant 20C3M/picntrl demand in CMIP3 (CMIP 64%, 35% ScenarioMIP).

forced "control" experiment identified core experiments between CMIP1 and CMIP2, with the present-day control (pdcntrl, ~1995 CMIP1) evolving to include a pre-industrial control (picntrl, ~1850-1860 CMIP2) in subsequent phases. When assessing the "control" experiments, the nomenclature changed a little, with picntrl (before CMIP5) and piControl referring to the same experimental protocols, noting differing "pre-industrial" climatological fixed forcings were used across phases (see subsection 5.1). Idealized experiments were also incorporated in CMIP2, with the 1% compounding (1pctCO2) first included and subsequently identified as 1pctto2x and 1pctto4x (before CMIP5), returning to the single 1pctCO2 identity in CMIP5 onward, and with differing simulation lengths across contributing models (2x 70 years, and 4x 140 years). The historical experiment, with transient time-evolving forcings included, was first defined in CMIP3, identified as 20C3M (climate of the twentieth century, ~1860-1999). This is subsequently the historical experiment (CMIP5 and onward) and was extended to include additional forcing coverage (CMIP5: 1850-2011; CMIP6: 1850-2014). For further details and comparisons, see Table A1, and to visualize the experiment growth over phases, see Figure 1.

## Appendix B:  MIP variable request, standard output and data growth

The information presented in Table 1 row "Standard output variables/Tables" was collated from numerous live and archived resources available from 1991 through to the CMIP6 CMOR Table files that are still being used today. The earliest resources were published in written form, the AMIP Newsletters (e.g., Gates, 1991), and subsequently, became available on the PCMDI website for AMIP2, CMIP1 and CMIP2/2+ phases. CMIP3 marked a step change, with the development of the CMOR1 software



**Table A1.** MIP Experiments AMIP1 (1991) through CMIP6

| MIP Phase | Citation/Year | Experiment(s) | URL/DOI |
|---|---|---|---|
| AMIP1 | Gates (1991) | **AMIP:** amip | 10.5281/zenodo.12109765; https://web.archive.org/web/19970524094021/http://www-pcmdi.llnl.gov/amip/ |
| AMIP2 | Gleckler (1999) | **AMIP:** amip | 10.5281/zenodo.12188729; https://web.archive.org/web/19970524094021/http://www-pcmdi.llnl.gov/amip/ |
| CMIP1 | Meehl (1995) | **CMIP:** pdcntrl | https://web.archive.org/web/19970824235843/http://www-pcmdi.llnl.gov/cmip/Cmip.htm |
| CMIP2 | Meehl et al. (1997) | **CMIP:** pdcntrl, picntrl, 1pctCO2 | https://web.archive.org/web/19970825000210/http://www-pcmdi.llnl.gov/cmip/announ.htm |
| CMIP3 | Doutriaux & Taylor, 2005; Meehl et al. (2007a) | **CMIP:** 1pctto2x, 1pctto4x, 20C3M, amip, pdcntrl, picntrl; **CFMIP:** 2xco2, slabcntl; **ScenarioMIP:** commit, SRESA1B, SRESA2, SRESB1 | https://pcmdi.llnl.gov/mips/cmip3/experiment.html#Experiments |
| CMIP5 | Taylor et al. (2011, 2012b); Doutriaux & Taylor, 2013 | **CMIP:** 1pctCO2, abrupt4xCO2, amip, historical, piControl; **C4MIP:** esmControl, esmFdbk1, esmFdbk2, esmFixClim1, esmFixClim2, esmHistorical, esmrcp85; **CFMIP:** amip4K, amip4xCO2, amipFuture, aqua4K, aqua4xCO2, aquaControl, sst2030 **DCPP:** decadalXXXX, noVolcXXXX, volcIn2010; **DAMIP:** historicalExt, historicalGHG, historicalMisc, historicalNat; **PMIP:** lgm, midHolocene, past1000; **RFMIP:** sstClim, sstClim4xCO2, sstClimAerosol, sstClimSulfate; **ScenarioMIP:** rcp26, rcp45, rcp60, rcp85 | https://pcmdi.llnl.gov/mips/cmip5/experiment_design.html |
| CMIP6 | Eyring et al. (2016a); Durack et al. (2024) | ~190 (2016) to 322 (2024) see CMIP6_CVs; for MIPs contributing to the phase see Table 2 | 10.5281/zenodo.12197150; https://github.com/WCRP-CMIP/CMIP6_CVs; https://wcrp-cmip.github.io/CMIP6_CVs/docs/CMIP6_experiment_id.html |

Notes: For CMIP3 and CMIP5, no "Endorsed-MIPs" were identified, rather experiments were included without recognizing the community that defined these experiments. To attempt to provide connectivity across CMIP phases, CMIP6-era MIP identities (see Table 2, e.g., CFMIP, ScenarioMIP) have been retrofitted back to prior phase experiments which led to corresponding experiments in CMIP6. In CMIP5, NOAA-GFDL submitted GFDL-HIRAM-C180 simulations for the CFMIP-motivated sst2090 and sst2090rcp45 experiments (as these were single model experiments they are not listed above).





**Figure B1.** Data growth across all MIP phases, AMIP1 1989 through CMIP6 today. This figure is a visual representation of tabulated data volumes in Table 1. Note a y-axis log-scale, capturing data growth from gigabytes/$10^9$ bytes for AMIP1 and CMIP1 through to the tens of petabyte/$10^{15}$ bytes scale in CMIP6.

(Taylor et al., 2006) and CMIP3 Standard Output defined by the more complete CMIP3-CMOR-Tables (Doutriaux and Taylor, 2005). CMIP5 continued this trend with CMOR2 (Doutriaux and Taylor, 2011) and the CMIP5-CMOR-Tables (Doutriaux and Taylor, 2013). For CMIP6, project expansion to include 22 MIPs and 322 experiments (Table 2, Table 1 respectively) required 1135 the development of a dedicated CMIP6 Data Request (see subsection 4.3) along with the parallel development of CMOR3 (Mauzey et al., 2024) and the CMIP6-CMOR-Tables (Nadeau et al., 2017). For each tabulated value, superscript-identified links provide live connections to these sources, see Table B1.



**Table B1.** MIP variable request and standard output AMIP1 (1991) to CMIP6

| Table 1 superscript | MIP Phase | Variable Count | CMOR version | Citation/Year | URL/DOI |
|---|---|---|---|---|---|
| 1 | AMIP1 | 32 | ∼ | Gates (1991) | 10.5281/zenodo.12109765; https://pcmdi.llnl.gov/mips/amip/OUTPUT/WGNEDIAGS/index.html |
| 2 | AMIP2 | 114 | ∼ | 1998 | https://pcmdi.llnl.gov/mips/amip/OUTPUT/AMIP2/outlist.html |
| 3 | CMIP1 | 23 | ∼ | 1997 | https://web.archive.org/web/19970824233750/http://www-pcmdi.llnl.gov/cmip/diagsub.html |
| 4 | CMIP2 | 28 | ∼ | 1997 | https://pcmdi.llnl.gov/mips/cmip2/ |
| 5 | CMIP3 | 143[†] | 1.0 | Doutriaux & Taylor, 2005 | 10.5281/zenodo.12792173; https://github.com/PCMDI/cmip3-cmor-tables |
| 6 | CMIP5 | 986 | 2.0 | Doutriaux & Taylor, 2013 | 10.5281/zenodo.12792191; https://github.com/PCMDI/cmip5-cmor-tables |
| 7 | CMIP6 | 2062 | 3.0 | Nadeau et al., 2018 | 10.5281/zenodo.597650; https://github.com/PCMDI/cmip6-cmor-tables |
| ∼ | CFMIP1 | 149 | 1.0 | ∼ | https://github.com/PCMDI/cfmip1-cmor-tables |
| ∼ | C-LAMP1 | 88 | 1.0 | ∼ | https://github.com/PCMDI/c-lamp1-cmor-tables |
| ∼ | IAEMIP1 | 146 | 1.0 | ∼ | https://github.com/PCMDI/iaemip1-cmor-tables |
| ∼ | CORDEX (CMIP5) | 207 | 2.0 | ∼ | https://github.com/PCMDI/cordex-cmor-tables |
| ∼ | GEOMIP | 1142 | 2.0 | ∼ | https://github.com/PCMDI/geomip-cmor-tables |
| ∼ | LUCID | 979 | 2.0 | ∼ | https://github.com/PCMDI/lucid-cmor-tables |
| ∼ | PMIP3 | 810 | 2.0 | ∼ | https://github.com/PCMDI/pmip3-cmor-tables |
| ∼ | CORDEX-CMIP6 | 565 | 3.0 | Gutowski Jr. et al. (2016) | https://github.com/WCRP-CORDEX/cordex-cmip6-cmor-tables |

Notes: [†]In the CMIP3 phase, the total defined A5 table variables were 4, both adjusted/instantaneous shortwave forcing and its clearsky equivalent. This differs from the 223 identities in the table, which identified similar quantities (either top of atmosphere or tropopause, and variations due to unique forcing, e.g., all greenhouse gases, carbon dioxide only, total sulphate aerosol, direct effect only of sulphate aerosol, indirect effect only of sulphate aerosols, "black carbon", ozone, tropospheric ozone only, stratospheric ozone only, vegetation and other land surface changes, all anthropogenic factors, inclusive, volcanic aerosols, solar constant changes, all-natural factors inclusive.



## Appendix C: MIP errata

Tabulated entries of CMIP errata based on phase, see Table C1.

**Table C1.** MIP Errata CMIP3 (2004) to CMIP6

| MIP Phase | Errata count (reporting period) | URL/DOI |
|:---:|:---:|:---:|
| CMIP3 | 122 (2004-2011) | http://web.archive.org/web/20150906073117/https://esg.llnl.gov:8443/about/errata.do |
| CMIP5 | 84 (2012-2015) | https://pcmdi.llnl.gov/mips/cmip5/errata.html |
| CMIP6 | 462 (2018-2024) | https://errata.ipsl.fr |

Notes: All values are tabulated from archived or live webpages as of $28^{th}$ November, 2024.

## Appendix D: CMIP6 data preparation tools

During MIP phases, several software tools have been developed and updated to meet augmented phase requirements. These tools build on the MIP nomenclature and digital formats that are now standard. Some prominent packages are tabulated below (see Table D1).

## Appendix E: Acronyms

Over the five decades of MIPs, many acronyms and identifiers have been developed and bled into standard nomenclature. Some of these are used repeatedly throughout the text, and we tabulate entries here (see Table E1). Additional identifiers used to describe CMIP6 Community MIPs are detailed in Table 2.

*Author contributions.* P.J.D. outlined the content, completed the initial outline, and shared responsibility for writing the manuscript. K.E.T. assisted in expanding the outline and shared responsibility for writing the manuscript. P.J.G. provided useful feedback on early drafts, which reformulated the outline. G.A.M provided useful context and feedback on section 2 and section 3. B.N.L. provided useful context and feedback on subsection 4.1, subsection 4.4, and subsection 4.7. C.C. provided useful context and feedback on section 2. R.J.S. provided useful context and feedback on section 2. G.L., A.B-N, and S.D. provided useful context and feedback on subsection 4.9 and subsection 4.11. M.S. provided useful context and feedback on subsection 4.5 and subsection 4.8. J.G. provided useful context and feedback on subsection 4.1. M.J. provided useful context and feedback on subsection 4.3. All authors contributed to the final version of the manuscript.

*Competing interests.* All authors declare no competing or conflicting interests. M.S. is a Topic Editor for Geoscientific Model Development.



**Table D1.** Software packages developed to aid MIP dataset production (non-exhaustive list)

| Software name and version | Software Description | URL | Citation | DOI |
|---|---|---|---|---|
| CMOR 1.0 | The Climate Model Output Rewriter | https://cmor.llnl.gov/archive/cmor1; https://github.com/PCMDI/cmor | Taylor et al. (2006) | 10.5281/ zen-odo.12690071 |
| CMOR 2.0 | The Climate Model Output Rewriter | https://cmor.llnl.gov; https://github.com/PCMDI/cmor | Doutriaux and Taylor (2011) | 10.5281/ zen-odo.12690366 |
| CMOR 3.0 | The Climate Model Output Rewriter | https://cmor.llnl.gov; https://github.com/PCMDI/cmor | Doutriaux et al. (2024) | 10.5281/ zen-odo.592733 |
| XIOS | Xml IO Server | http://forge.ipsl.jussieu.fr/ioserver/chrome/site/XIOS_DOC | | |
| ECE2CMOR3 | EC-Earth to CMOR | https://github.com/EC-Earth/ece2cmor3; https://github.com/EC-Earth/cmor-fixer | | 10.5281/ zen-odo.1051094 |
| CDO CMOR | Climate Data Operators to CMOR | https://code.mpimet.mpg.de/projects/cdo/wiki/CDO_CMOR_ Operator | | |
| NORESM2CMOR | NorESM to CMOR | https://github.com/NorESMhub/noresm2cmor | | |
| CCLM2CMOR | COSMO-CLM to CMOR | https://github.com/C2SM-RCM/CCLM2CMOR; https://github.com/ssilje/CMOR | | |
| FGOALS-g-cmor | FGOALS-g to CMOR | https://github.com/dongli/FGOALS-g-cmor | | |
| PRIMAVERA | HadGEM to CMOR | https://github.com/goord/cmor | | |
| E3SM_To_CMIP | DoE-E3SM to CMOR | https://github.com/E3SM-Project/e3sm_to_cmip | | |
| ACCESS MOPPeR | A Model Output Post-Processor for the ACCESS climate model | https://access-mopper.readthedocs.io/ | | |



**Table E1.** Acronyms used in this manuscript

| Acronym | Expansion and additional information |
|---------|-------------------------------------|
| 20C3M | CMIP 20th Century Climate in Coupled Models pilot project (now known as the CMIP historical experiment) |
| AGCI | US Aspen Global Change Institute; https://www.agci.org |
| AGCM | Atmospheric General Circulation Model |
| AMIP | Atmospheric Model Intercomparison Project (also AMIP1 and AMIP2) |
| ANL | US Argonne National Laboratory; https://www.anl.gov |
| AOGCM | Atmospheric and Ocean General/Global Circulation Model |
| AR4 | IPCC Fourth Assessment Report, 2007; https://www.ipcc.ch/report/ar4/wg1 |
| AR5 | IPCC Fifth Assessment Report, 2013; https://www.ipcc.ch/report/ar5/wg1 |
| AR6 | IPCC Sixth Assessment Report, 2021; https://www.ipcc.ch/report/ar6/wg1 |
| BADC/CEDA | UK British Atmospheric Data Centre (now CEDA; https://www.ceda.ac.uk) |
| C-LAMP | Carbon-Land Model Intercomparison Project; (now ILAMB; https://www.ilamb.org) |
| C4MIP | Coupled Climate-Carbon Cycle MIP; https://c4mip.net |
| CCSM | NCAR Community Climate System Model; https://www.cesm.ucar.edu/models/ccsm |
| CEDA | UK Centre for Environmental Data Analysis; https://www.ceda.ac.uk |
| CF | NetCDF Climate and Forecast Metadata Conventions; https://cfconventions.org/ |
| CFMIP | Cloud Feedbacks MIP (Also CFMIP1, CFMIP2, and CFMIP3; https://www.cfmip.org) |
| CIESIN | US Center for Integrated Earth System Information (Columbia University); https://sedac.ciesin.columbia.edu/ddc/ |
| CLIVAR | WCRP Climate Variability and Predictability Core Project; https://www.clivar.org |
| CMCC | Italian CMCC Foundation (Euro-Mediterranean Center on Climate Change); https://www.cmcc.it |
| CMIP | Coupled Model Intercomparison Project (also CMIP1, CMIP2, CMIP2+, CMIP3, CMIP5, and CMIP6) |
| CMIP-IPO | UK CMIP International Project Office; https://wcrp-cmip.org |
| CMOR | PCMDI Climate Model Output Rewriter (also CMOR1, CMOR2, and CMOR3; https://cmor.llnl.gov/) |
| COARDS | Cooperative Ocean/Atmosphere Research Data Service conventions |
| CRU | UK University of East Anglia, Climatic Research Unit; https://www.uea.ac.uk/groups-and-centres/climatic-research-unit |




**Table E1.** Acronyms used in this manuscript (continued)

| | |
|---|---|
| CV | Controlled Vocabulary |
| DandA | climate change Detection and Attribution research |
| DAMIP | Detection and Attribution MIP |
| DCPP | Decadal Climate Prediction Project (CMIP6 Community MIP, and WCRP ESMO Working Group; https://www.wcrp-climate.org/dcp-overview) |
| DECK | Diagnostic, Evaluation and Characterisation of Klima (Core CMIP experiment suite) |
| DoE | US Department of Energy; https://www.energy.gov |
| DOI | Digital Object Identifier; https://www.doi.org |
| DKRZ | German Climate Computing Center (Deutsches Klimarechenzentrum; https://www.dkrz.de/en) |
| DLR | German Deutsches Zentrum für Luft- und Raumfahrt, Institut für Physik der Atmosphäre; https://www.dlr.de/en |
| DRS | Data Reference Syntax |
| ECMWF | European Centre for Medium-range Weather Forecasts; https://www.ecmwf.int |
| ESA | European Space Agency; https://www.esa.int |
| ESG | US Earth System Grid (also ESG I, ESG II) |
| ESG-CET | US Earth System Grid Center for Enabling Technologies |
| ESGF | Earth System Grid Federation; https://esgf.llnl.gov |
| ESM | Earth System Model |
| ESMValTool | A community diagnostic and performance metrics tool for evaluation and analysis of Earth system Models; https://esmvaltool.org |
| ESMO | WCRP Earth System Modelling and Observations Core Project; https://www.wcrp-esmo.org |
| FANGIO | Feedback ANalysis of GCMs and In Observations project |
| FAR | IPCC First Assessment Report, 1990; https://www.ipcc.ch/report/ar1/wg1 |
| FMI | Finnish Meteorological Institute; https://en.ilmatieteenlaitos.fi |
| FTP | File Transfer Protocol (the standard protocol used to transfer files from a server to a client on a network; https://en.wikipedia.org/wiki/File_Transfer_Protocol |
| GridFTP | Extension of FTP for grid computing; https://www.globus.org/blog/gridftp-a-brief-history-of-fast-file-transfer |
| GAIM | Global Analysis Interpretation and Modeling Task Force (IGBP sub-group) |
| GARP | International Global Atmospheric Research Program (a precursor to the WCRP; 1967-1982) |



**Table E1.** Acronyms used in this manuscript (continued)

| GB | Gigabyte (one billion bytes, $10^9$) |
|---|---|
| GCM | General/Global Circulation Model |
| GDT | Gregory, Drach, and Tett conventions; (see Gregory et al., 1999) |
| GS | Google Scholar citation service; https://scholar.google.com |
| HTTP | HyperText Transfer Protocol (the default protocol underlying internet transactions; https://httpwg.org/specs |
| IAM | Integrated Assessment Model |
| IAV | Impacts, Adaptation and Vulnerability research |
| ICRCCM | Intercomparison of Radiation Codes used in Climate Models project |
| IGBP | International Geosphere-Biosphere Programme (closed in 2015; http://www.igbp.net |
| ILAMB | ORNL International Land Model Benchmarking; https://www.ilamb.org |
| IPCC | UN Intergovernmental Panel on Climate Change; https://www.ipcc.ch |
| IPCC DDC | IPCC Data Distribution Centre; https://www.ipcc-data.org |
| IPO | International Project Office (e.g., CMIP-IPO, https://wcrp-cmip.org/cmip-governance/project-office/ |
| IPSL | French Institut Pierre-Simon Laplace; https://www.ipsl.fr/en/home-en |
| IS-ENES | Infrastructure for the European Network for Earth System Modelling; https://is.enes.org |
| JSON | JavaScript Object Notation; text-based format for storing and exchanging data, both human-readable and machine-parseable; https://www.json.org |
| LANL | US Los Alamos National Laboratory; https://www.lanl.gov |
| LBNL | US Lawrence Berkeley National Laboratory; https://www.lbl.gov |
| LLNL | US Lawrence Livermore National Laboratory; https://www.llnl.gov |
| MIP | Model Intercomparison Project |
| NASA | US National Aeronautic and Space Administration; https://www.nasa.gov |
| NASA-JPL | US NASA Jet Propulsion Laboratory; https://www.jpl.nasa.gov |
| NCAR | US National Center for Atmospheric Research; https://ncar.ucar.edu |
| NCI | Australian National Computational Infrastructure; https://nci.org.au |
| NERSC | US Department of Energy Research Scientific Computing Center; https://www.nersc.gov |

**Table E1 continued overpage..**



**Table E1.** Acronyms used in this manuscript (continued)

| | |
|---|---|
| NOAA | US National Oceanic and Atmospheric Administration; https://www.noaa.gov |
| NOAA-NCEP | US NOAA National Centers for Environmental Prediction; https://www.weather.gov/ncep |
| NOAA-PMEL | US NOAA Pacific Marine Environmental Laboratory; https://www.pmel.noaa.gov |
| OCMIP | Ocean Carbon-Cycle Model Intercomparison Project; https://www.wcrp-climate.org/modelling-wgcm-mip-catalogue/modelling-wgcm-mips-2/267-modelling-wgcm-catalogue-ocmip |
| ORNL | US Oak Ridge National Laboratory; https://www.ornl.gov |
| PB | Petabyte (one quadrillion bytes, $10^{15}$) |
| PMIP | CMIP Paleoclimate MIP (also PMIP1, PMIP2, PMIP3, and PMIP4; https://pmip.lsce.ipsl.fr) |
| PMP | PCMDI Metrics Package; https://pcmdi.github.io/pcmdi_metrics |
| PCMDI | Program for Climate Model Diagnosis and Intercomparison, LLNL; https://pcmdi.llnl.gov |
| P-DRS | PCMDI Data Retrieval and Storage software library (digital format; Drach and Mobley, 1995) |
| RCP | Representative Concentration Pathways scenarios (Circa CMIP5) |
| SAR | IPCC Second Assessment Report, 1995; https://www.ipcc.ch/report/ar2/wg1 |
| SGGCM | WCRP Steering Group on Global Coupled Models (a precursor to WGCM) |
| SLCF | short-lived climate forcers |
| SOLR | Apache SOLR, open-source enterprise-search database; https://solr.apache.org |
| SPECTRE | Spectral Radiance Experiment |
| SRES | IPCC Special Report on Emission Scenarios (Circa CMIP3) |
| SST | Sea Surface Temperature |
| TAR | IPCC Third Assessment Report, 2001; https://www.ipcc.ch/report/ar3/wg1 |
| TB | Terabyte (one trillion bytes, $10^{12}$) |
| Unidata | US Unidata Program Center (University Corporation of Atmospheric Research); https://www.unidata.ucar.edu |
| WoS | Clarivate Web of Science Core Collection citation service; https://www.webofscience.com/wos/woscc |
| WCRP | World Climate Research Programme; https://www.wcrp-climate.org |
| WDAC | WCRP Data Advisory Council (2011-2020; https://www.wcrp-climate.org/data-wdac |
| WG1 | IPCC Working Group I (the Physical Science Basis); https://www.ipcc.ch/working-group/wg1 |



**Table E1.** Acronyms used in this manuscript (continued)

| | |
|---|---|
| WG2 | IPCC Working Group II (Impacts, Adaptation, and Vulnerability); https://www.ipcc.ch/working-group/wg2 |
| WG3 | IPCC Working Group III (Mitigation of Climate Change); https://www.ipcc.ch/working-group/wg3 |
| WGCM | WCRP Working Group on Coupled Modelling; https://www.wcrp-climate.org/ipo-esmo-groups/modelling-wgcm |
| WGNE | WCRP Working Group on Numerical Experimentation; https://wgne.net |
| WIP | WCRP WGCM Infrastructure Panel; https://www.wcrp-climate.org/wgcm-cmip/wip |
| WMGHG | well-mixed greenhouse gases |
| UN | United Nations; https://www.un.org/en |
| US | United States |
| VIACS AB | Vulnerability, Impacts, Adaptation and Climate Services Advisory Board; https://viacsab.gerics.de |

*Disclaimer.* The views and opinions expressed in this document do not necessarily state or reflect those of the United States Government, the Department of Energy (DoE), or Lawrence Livermore National Laboratory (LLNL). They shall not be used for advertising or product endorsement purposes.

*Acknowledgements.* This article is dedicated to W. Lawrence "Larry" Gates, the first PCMDI Director, whose vision, leadership, and skill
in building community engagement and consensus garnered acceptance for open and systematic climate model analysis and led to the transformative impact of CMIP.

We acknowledge all past and present members of the WGCM Infrastructure Panel (WIP), CMIP Panel, and Earth System Grid Federation (ESGF) and their precedent organizations. Without the collective efforts of these teams, the progress that CMIP embodies would not have been possible. The authors listed in this article represent a tiny subset of the project contributors over time.

We acknowledge all MIP contributing modelling groups, the World Climate Research Programme's (WCRP) Working Group on Coupled Modelling (WGCM) and its WGCM Infrastructure (WIP), and the CMIP Panels for their leadership in defining and delivering multiple CMIP and preceding AMIP phases. The simulation contributions, which now embody petabytes of the latest generation of historical and future climate data, are the tangible substance of the CMIP project across phases.

We acknowledge the Program for Climate Model Diagnosis and Intercomparison (PCMDI, US), the Centre for Environmental Data
Analysis (CEDA, UK), the Infrastructure for the European Network for Earth System Modelling (IS-ENES), the German Climate Computing Centre (DKRZ, Germany), the Institute Pierre-Simon Laplace (IPSL, France), Centro Euro-Mediterraneo sui Cambiamenti Climatici (CMCC, Italy), the Earth System Grid Federation (ESGF) and Earth System Grid (ESG) contributing organizations and many other infrastructure providers for developing the ecosystem that delivered MIP data across multiple phases.

We gratefully acknowledge Vaishali Naik (NOAA Geophysical Fluid Dynamics Laboratory, Princeton, USA) for helping us better under-
1175 stand the climate forcing experience of modelling groups through the CMIP3, and CMIP5 phases.



We thank numerous colleagues, collaborators, and past AMIP, CMIP, and ESGF contributors for their valuable input and feedback.

The work of P.J.D., K.E.T., P.J.G., C.C., S.K.A., D.C.B, J.L., C.F.M., J.P. and G.L.P. from Lawrence Livermore National Laboratory (LLNL) is supported by the Regional and Global Model Analysis (RGMA) program area under the Earth and Environmental System Modeling (EESM) program within the Earth and Environmental Systems Sciences Division (EESSD) of the United States Department of Energy's (DoE) Office of Science (OSTI). This work was performed under the auspices of the US DoE by LLNL under contract DE-AC52-07NA27344. LLNL IM Release: LLNL-JRNL-871359.

The work of G.A.M. is supported by the Regional and Global Model Analysis (RGMA) program area under the Earth and Environmental System Modeling Program (EESM) of the US Department of Energy's Office of Biological and Environmental Research (BER) under Award Number DE-SC0022070. This work was also supported by the National Center for Atmospheric Research, a major facility sponsored by the National Science Foundation (NSF) under Cooperative Agreement No. 1852977.

V.E's work is supported by the European Research Council (ERC) Synergy Grant 'Understanding and Modeling the Earth System with Machine Learning' (USMILE) under the Horizon 2020 Research and Innovation programme (grant agreement no. 855187) and the Deutsche Forschungsgemeinschaft (DFG, German Research Foundation) through the Gottfried Wilhelm Leibniz Prize awarded to V.E. (reference no. EY 22/2-1).

D.E. and E.O's work at the CMIP-IPO is hosted by the European Space Agency, with staff provided on contract by HE Space Operations Ltd.



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
