# Peer review of "The Coupled Model Intercomparison Project (CMIP): Reviewing project history, evolution, infrastructure and implementation"

_EGUsphere, 2024_

## Referee Comment (RC1)

**Review** of 'The Coupled Model Intercomparison Project (CMIP): Reviewing project history, evolution, infrastructure and implementation' by Durack etc al (2024)

Catherine Senior
6/2/2025

**General**

I found this paper to be a very valuable addition to the literature on CMIP. It details the long evolution of the project bringing together historical beginnings, timelines and importantly many aspects of the supporting infrastructure that has not previously been detailed for the wider audience. I think this gives the reader more context and understanding of the immense work needed to support the increasingly large and diverse users of co-ordinated climate model projections.

I have provided some more detailed comments below, but in general I thought the paper was well balanced and I didn't think statements about the impact of CMIP were hyperbolic - noting specifically the discussion (Section 6: CMIP Impact) of how difficult/unsatisfactory it is to support any statements of impact through use of citations and downloads numbers. Having said this, I did feel that some of the statements in this section and the Abstract and Summary (Section 8) could perhaps be better formulated to avoid accusations of hyperbole. I have made some suggestions in the detailed comments below.

In the opening sections (e.g. Section 2), I felt that there was too much of a US focus and parallel initiatives in Europe (e.g IS-ENES) and wider world-wide activities of modelling groups (e.g. NICAM in Japan) were not mentioned. Some of these come later in the paper, but at least a cross reference at the start would help to recognise the growing International contribution to the project in time.

In Section 7, perhaps more could be said about the thinking and planning on the design/size of CMIP7. I think this would help the reader to understand that those involved don't see CMIP as continually growing and recognise the pressures on the modelling centres delivering large numbers of experiments. I also think a bit more discussion on the move to operationalise some of the policy-relevant simulations (the Fast Track) and thus provide a separation in time between the Service and Research aspects would highlight the efforts to bridge CMIP to ideas arising in the community (e.g. EVE; Jakob et al, 2023; Stevens 2024).

**Detailed comments**

L7: Surely the provision of future scenarios and the policy decisions made from these is worth mentioning as much as the capability to quantify and attribute observed changes.

L96-97 and L359-362: Of the 12 models submitted to AMIP1 - did all groups use the NERSC HPC to run the simulations - even those in Europe?

L367: Again, this seems a bit US focussed. CMIP data was certainly on the IPCC DDC before CMIP5 and many will have accessed this.

L583: I dont understand the sentence 'Still, it is disappointing that two or fewer groups provided some 562 CMIP6 requested variables, and no one provided 185'

L595: In CMIP6 there was a very useful set of variables designated as IPCC priority that a lot of modelling centres used to decide what order to deliver the data.

L698/Section 4.6: I think it would be useful to reference this section earlier (e.g. in Section 2) to avoid the opening context being so US centric.

L730: Might be worth mentioning that an improved model documentation process is firmly in the sights of CMIP7 planning.

L1040: Although I agree with this statement, I am aware that others don't and I wonder if perhaps the statement 'However, without CMIP, the IPCC assessments could not have been possible' could be toned down to something less controversial like ' However, CMIP simulations have underpinned every IPCC assessment'?

L1070 and L1101: Would be worth saying that the Fast Track was at least partly motivated by the community feedback about the large number of experiments and also the desire to separate operational and research parts of CMIP. I think it could be made clearer that the CMIP panel is absolutely responding to what the community has said and also looking to bridge with other initiatives.

L1082: Instead of 'CMIP has generated profound scientific insights ….' How about 'CMIP has enabled (or supported?) profound scientific insights…'?  Might sound less grandiose..?

L1083: Could change 'It has improved climate prediction, provided quantified insights ….' To 'It has provided rigour and robustness to climate prediction, provided quantified insights …'

L1090-L1094: I think it would be helpful to expand a bit here on efforts by the CMIP panel to bridge to community ideas around operationalisation and high-resolution (e.g. EVE). The key here is that none of these other efforts are in a position to support the community now and for the next 5+ years. Only CMIP can do this.